# Theory for the optimal detection of time-varying signals in cellular sensing systems

**Giulia Malaguti, Pieter Rein ten Wolde***

AMOLF, Science Park, Amsterdam, Netherlands

**Abstract** Living cells often need to measure chemical concentrations that vary in time, yet how accurately they can do so is poorly understood. Here, we present a theory that fully specifies, without any adjustable parameters, the optimal design of a canonical sensing system in terms of two elementary design principles: (1) there exists an optimal integration time, which is determined by the input statistics and the number of receptors; and (2) in the optimally designed system, the number of independent concentration measurements as set by the number of receptors and the optimal integration time equals the number of readout molecules that store these measurements and equals the work to store these measurements reliably; no resource is then in excess and hence wasted. Applying our theory to the *Escherichia coli* chemotaxis system indicates that its integration time is not only optimal for sensing shallow gradients but also necessary to enable navigation in these gradients.

**\*For correspondence:**
tenwolde@amolf.nl

**Competing interests:** The authors declare that no competing interests exist.

## Introduction

Living cells continually have to respond and adapt to changes in their environment. They often do so on a timescale that is comparable to that of the environmental variations. Examples are cells that during their development differentiate in response to time-varying morphogen gradients (*Durrieu et al., 2018*) or cells that navigate through their environment (*Tostevin and ten Wolde, 2009*; *Sartori and Tu, 2011*; *Long et al., 2016*). These cells shape, via their movement, the statistics of the input signal, such that the timescale of the input fluctuations becomes comparable to that of the response. In all these cases, it is important to understand how accurately the cell can estimate chemical concentrations that vary in time.

Cells measure chemical concentrations via receptors on their surface. These measurements are inevitably corrupted by the stochastic arrival of the ligand molecules by diffusion and by the stochastic binding of the ligand to the receptor. Wiener and Kolmogorov (*Extrapolation, 1950*; *Kolmogorov, 1992*) and *Kalman, 1960* have developed theories for the optimal strategy to estimate signals in the presence of noise. Their filtering theories have been employed widely in engineering, and in recent years they have also been applied to cell signaling. They have been used to show that time integration can improve the sensing of time-varying signals by reducing receptor noise, although it cannot remove this input noise completely because of signal distortion (*Andrews et al., 2006*; *Hinczewski and Thirumalai, 2014*; *Becker et al., 2015*). It has been shown that circadian systems can adapt their response to the statistics of the input signal, as predicted by Kalman filtering theory (*Husain et al., 2019*). Moreover, Wiener–Kolmogorov filtering theory has been employed to derive the optimal topology of the cellular network depending on the statistics of the input signal (*Becker et al., 2015*). Negative feedback and incoherent feedforward, which are common motifs in cell signaling (*Alon, 2007*), make it possible to predict future signal values via signal extrapolation, which is useful when the past signal contains information about the future in addition to the current signal (*Becker et al., 2015*).

The precision of sensing depends not only on the topology of the cellular sensing network but also on the resources required to build and operate it. Receptors and time are needed to take the

concentration measurements (*Berg and Purcell, 1977*), downstream molecules are necessary to store the ligand-binding states of the receptor in the past, and energy is required to store these states reliably (*Govern and Ten Wolde, 2014a*). Many studies have addressed the question how receptors and time limit the precision of sensing static concentrations that do not vary on the time-scale of cellular response (*Berg and Purcell, 1977*; *Bialek and Setayeshgar, 2005*; *Wang et al., 2007*; *Rappel and Levine, 2008*; *Endres and Wingreen, 2009*; *Hu et al., 2010*; *Mora and Wingreen, 2010*; *Govern and Ten Wolde, 2012*; *Mehta and Schwab, 2012*; *Govern and Ten Wolde, 2014a*; *Govern and Ten Wolde, 2014b*; *Kaizu et al., 2014*; *Ten Wolde et al., 2016*; *Mugler et al., 2016*; *Fancher and Mugler, 2017*). In addition, progress has been made in understanding how the number of readout molecules and energy set the precision of sensing static signals (*Mehta and Schwab, 2012*; *Govern and Ten Wolde, 2014a*; *Govern and Ten Wolde, 2014b*). Yet, what the resource requirements for sensing time-varying signals are is a wide open question. In particular, it is not known how the number of receptor and readout molecules, time, and power required to maintain a desired sensing precision depend on the strength and the timescale of the input fluctuations.

In this article, we present a theory for the optimal design of cellular sensing systems as set by resource constraints and the dynamics of the input signal. The theory applies to one of the most common motifs in cell signaling, a receptor that drives a push–pull network, which consists of a cycle of protein activation and deactivation (*Goldbeter and Koshland, 1981*, see *Figure 1*). These systems are omnipresent in prokaryotic and eukaryotic cells (*Alon, 2007*). Examples are GTPase cycles, as in the Ras system, phosphorylation cycles, as in MAPK cascades, and two-component systems like the chemotaxis system of *Escherichia coli*. Push–pull networks constitute a simple exponential filter (*Hinczewski and Thirumalai, 2014*; *Becker et al., 2015*), in which the current output depends on the current and past input (with past input values contributing to the output with a weight that decays exponentially with time back into the past). Wiener–Kolmogorov filtering theory (*Extrapolation, 1950*; *Kolmogorov, 1992*) shows that these networks are optimal for estimating signals that are memoryless (*Becker et al., 2015*), meaning that the past input does not contain information that

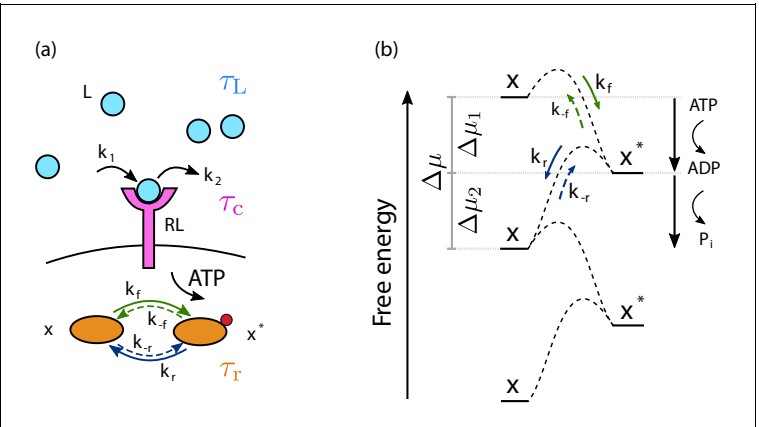

**Figure 1.** The cell signaling network. (a) The time-varying ligand concentration is modeled as a memoryless (Markovian) signal with mean $\overline{L}$, variance $\sigma_L^2$, and correlation time $\tau_L = \lambda^{-1}$. A free ligand molecule L (light blue circle) can bind at rate $k_1$ to a free receptor R (magenta protein) on the cell membrane (black line), forming the complex RL, and unbind at rate $k_2$ from RL. The correlation time of the receptor state is $\tau_c$. The complex RL catalyzes the phosphorylation reaction, driven by adenosine triphosphate (ATP) conversion, of a downstream readout from the unphosphorylated (inactive) state $x$ to the phosphorylated (active) state $x^*$, with rate $k_f$. The phosphorylated readout then spontaneously decays to the $x$ state with rate $k_r$. Microscopic reverse reactions of each signaling pathway are represented by dashed arrows. The relaxation time of the push–pull network is $\tau_r$. (b) Free-energy landscape of a readout molecule across the activation/deactivation reactions. Fuel turnover, provided by ATP conversion, drives the activation (phosphorylation) reaction characterized by the forward rate $k_f$ and its microscopic reverse rate $k_{-f}$ (green arrows). Associated with this activation reaction is a free-energy drop $\Delta\mu_1 = \log\frac{k_f \overline{x}}{k_{-f} \overline{x^*}}$. The deactivation pathway corresponds to the spontaneous release of the inorganic phosphate; it is characterized by the rate $k_r$ and its microscopic reverse $k_{-r}$ (blue arrows) and corresponds to a free-energy drop $\Delta\mu_2 = \log\frac{k_r \overline{x^*}}{k_{-r} \overline{x}}$.

is not already present in the current input. These networks are useful because they act as low-pass filters, removing the high-frequency receptor–ligand-binding noise (*Andrews et al., 2006*; *Hinczewski and Thirumalai, 2014*; *Becker et al., 2015*). Push–pull networks thus enable the cell to employ the mechanism of time integration, in which the cell infers the concentration not from the instantaneous number of ligand-bound receptors, but rather from the average receptor occupancy over an integration time (*Berg and Purcell, 1977*). Our theory gives a unified description in terms of all the cellular resources, protein copies, time, and energy, that are necessary to implement this mechanism of time integration. It does not address the sensing strategy of maximum-likelihood estimation (*Endres and Wingreen, 2009*; *Mora and Wingreen, 2010*; *Lang et al., 2014*; *Hartich and Seifert, 2016*; *Ten Wolde et al., 2016*) or Bayesian filtering (*Mora and Nemenman, 2019*).

While filtering theories are powerful tools for predicting the optimal topology and response dynamics of the cellular sensing network (*Andrews et al., 2006*; *Hinczewski and Thirumalai, 2014*; *Becker et al., 2015*), they do not naturally reveal the resource requirements for sensing. Our theory therefore employs the sampling framework of *Govern and Ten Wolde, 2014a* and extends it here to time-varying signals. This framework is based on the observation that the cell estimates the current ligand concentration not from the current number of active readout molecules directly, but rather via the receptor: the cell uses its push–pull network to estimate the receptor occupancy from which the ligand concentration is then inferred (see *Figure 2*). To elucidate the resource requirements for time integration, the push–pull network is viewed as a device that employs the mechanism of time integration by discretely sampling, rather than continuously integrating, the state of the receptor via collisions of the readout molecules with the receptor proteins (see *Figure 2*). During each collision, the ligand-binding state of the receptor protein is copied into the activation state of

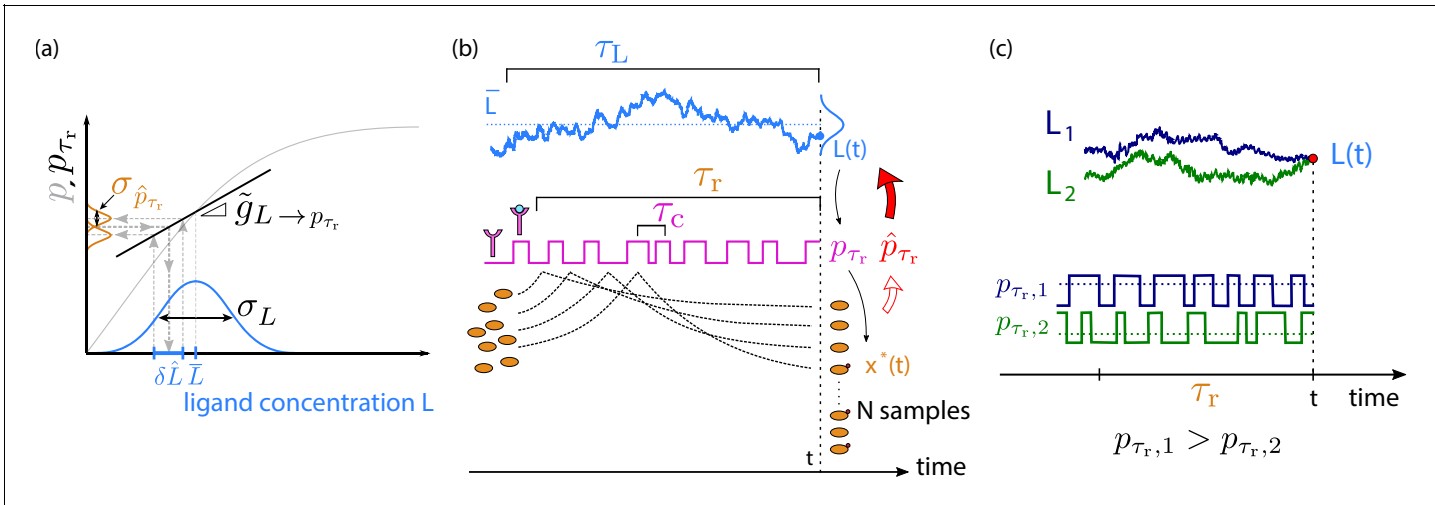

**Figure 2.** The precision of estimating a time-varying ligand concentration $L$. (a) The cell estimates the current ligand concentration $L = L(t)$ by estimating the average receptor occupancy $p_{\tau_r}$ over the past integration time $\tau_r$ and by inverting the dynamic input–output relation $p_{\tau_r}(L)$ (black solid line). The latter describes the mapping between the current concentration $L(t)$ of the time-varying signal and the average receptor occupancy $p_{\tau_r}$ over the past $\tau_r$, see also (b); it depends on the timescale $\tau_L$ of the input signal and hence differs from the conventional static input–output relation $p(L_s)$, which describes the mapping between the average receptor occupancy and a static ligand concentration $L_s$ that does not vary in time (gray solid line). The squared error in the estimate of the concentration $(\delta \hat{L})^2 = \sigma^2_{\hat{p}_{\tau_r}} / \tilde{g}^2_{L \to p_{\tau_r}}$ depends on the variance $\sigma^2_{\hat{p}_{\tau_r}}$ in the estimate of the average receptor occupancy $\hat{p}_{\tau_r}$ and the dynamic gain $\tilde{g}_{L \to p_{\tau_r}}$, which is the slope of $p_{\tau_r}(L)$. Only in the limit $\tau_c, \tau_r \ll \tau_L$, does $p_{\tau_r}(L)$ reduce to (the linearized form of) $p(L_s)$ and does the dynamic gain $\tilde{g}_{L \to p_{\tau_r}}$ become the static gain $g_{L \to p}$, which is the slope of $p(L_s)$ at the average ligand concentration $\bar{L}$. The input distribution, shown in blue, has width $\sigma_L$. (b) The average receptor occupancy $p_{\tau_r}$ over the past integration time $\tau_r$ is estimated via the downstream network, which is modeled as a device that discretely samples the ligand-binding state of the receptor via its readout molecules $x$ (*Govern and Ten Wolde, 2014a*); the fraction of modified readout molecules provides an estimate of $p_{\tau_r}$. The sensing error has two contributions (*Equation 6*): sampling error and dynamical error. The sampling error arises from the error in the estimate of $p_{\tau_r}$ that is due to the stochasticity of the sampling process; it depends on the number of samples, their independence, and their accuracy. (c) The dynamical error arises because the current ligand concentration $L(t)$ is estimated via the average receptor occupancy $p_{\tau_r}$ over the past integration time $\tau_r$: the latter depends on the ligand concentration in the past $\tau_r$, which will, in general, deviate from the current concentration. Two different input trajectories ($L_1$ in blue, $L_2$ in green) ending at time $t$ at the *same* value $L(t)$ (red dot) lead to different estimates of $L(t)$ due to their different average receptor occupancy ($p_{\tau_r,1} > p_{\tau_r,2}$) in the past $\tau_r$.

the readout molecule (*Ouldridge et al., 2017*). The readout molecules thus constitute samples of the receptor state, and the fraction of active readout molecules provides an estimate of the average receptor occupancy. The readout activation states have, however, a finite lifetime, which means that this is an estimate of the (running) average receptor occupancy over this lifetime, which indeed sets the receptor integration time $\tau_r$. The cell can estimate the current ligand concentration $L$ from this estimate of the average receptor occupancy $p_{\tau_r}$ over the past integration time $\tau_r$ because there is a unique one-to-one mapping between $p_{\tau_r}$ and $L$. This mapping $p_{\tau_r}(L)$ is the dynamic input–output relation and differs from the conventional static input–output relations used to describe the sensing of static concentrations that do not vary on the timescale of the response (*Berg and Purcell, 1977*; *Bialek and Setayeshgar, 2005*; *Kaizu et al., 2014*; *Ten Wolde et al., 2016*) in that it depends not only on the response time of the system but also on the dynamics of the input signal.

Our theory reveals that the sensing error can be decomposed into two terms, which each depend on collective variables that reveal the resource requirements for sensing. One term, the sampling error, describes the sensing error that arises from the finite accuracy by which the receptor occupancy is estimated. This error depends on the number of receptor samples, as set by the number of receptors, readout molecules, and the integration time; their independence, as given by the receptor-sampling interval and the timescale of the receptor–ligand-binding noise; and their reliability, as determined by how much the system is driven out of thermodynamic equilibrium via fuel turnover. The other term is the dynamical error and is determined by how much the concentration in the past integration time reflects the current concentration that the cell aims to estimate; it depends on the integration time and timescale of the input fluctuations.

Our theory gives a comprehensive view on the optimal design of a cellular sensing system. Firstly, it reveals that the resource allocation principle of *Govern and Ten Wolde, 2014a* can be generalized to time-varying signals. There exist three fundamental resource classes – receptors and their integration time, readout molecules, and power and integration time – which each fundamentally limit the accuracy of sensing; and, in an optimally designed system, each resource class is equally limiting so that none of them is in excess and thus wasted. However, in contrast to sensing static signals, time cannot be freely traded against the number of receptors and the power to achieve a desired sensing precision: there exists an optimal integration time that maximizes the sensing precision, which arises as a trade-off between the sampling error and dynamical error. Together with the resource allocation principle, it completely specifies, without any adjustable parameters, the optimal design of the system in terms of its resources protein copies, time, and energy.

Our theory also makes a number of specific predictions. The optimal integration time decreases as the number of receptors is increased because this allows for more instantaneous measurements. Moreover, the allocation principle reveals that when the input varies more rapidly both the number of receptors and the power must increase to maintain a desired sensing precision, while the number of readout molecules does not.

Finally, we apply our theory to the chemotaxis system of *E. coli*. This bacterium searches for food via a run-and-tumble strategy (*Berg and Brown, 1972*), yielding a fluctuating input signal. In small gradients, the timescale of these input fluctuations is set by the typical run time of the bacterium, which is on the order of a few seconds (*Berg and Brown, 1972*; *Taute et al., 2015*), while the strength of these fluctuations is determined by the steepness of the gradient. Interestingly, experiments have revealed that *E. coli* can sense extremely shallow gradients, with a length scale of approximately $10^4 \mu$m (*Shimizu et al., 2010*), raising the question how accurately *E. coli* can measure the concentration and whether this is sufficient to determine whether during a run it has changed, even in these shallow gradients. To measure the concentration, the chemotaxis system employs a push–pull network to filter out the high-frequency receptor–ligand-binding noise (*Sartori and Tu, 2011*). Applying our theory to this system predicts that the measured integration time, on the order of 100 ms (*Sourjik and Berg, 2002*), is not only sufficient to enable navigation in these shallow gradients but also necessary. This suggests that this system has evolved to optimally sense shallow concentration gradients.

## Results

### Theory: model

We consider a single cell that needs to sense a time-varying ligand concentration $L(t)$ (see *Figure 1a*). The ligand concentration dynamics is modeled as a stationary memoryless, or Markovian, signal specified by the mean (total) ligand concentration $\overline{L}$, the variance $\sigma_L^2$, and the correlation time $\tau_L = \lambda^{-1}$, which determines the timescale on which input fluctuations decay. It obeys Gaussian statistics (*Tostevin and ten Wolde, 2010*).

The concentration is measured via $R_T$ receptor proteins on the cell surface, which independently bind the ligand (*Ten Wolde et al., 2016*), $L + R \underset{k_2}{\overset{k_1}{\rightleftharpoons}} RL$. The correlation time of the receptor state, which is the timescale on which fluctuations in the number of ligand-bound receptors regresses to the mean, is given by $\tau_c = 1/(k_1\overline{L} + k_2)$ (*Berg and Purcell, 1977*; *Bialek and Setayeshgar, 2005*; *Kaizu et al., 2014*; *Ten Wolde et al., 2016*). It determines the timescale on which independent concentration measurements can be made.

The ligand-binding state of the receptor is read out via a push–pull network (*Goldbeter and Koshland, 1981*). The most common scheme is phosphorylation fueled by the hydrolysis of adenosine triphosphate (ATP) (see *Figure 1b*). The receptor, or an enzyme associated with it such as CheA in *E. coli*, catalyzes the modification of the readout, $x + RL + ATP \rightleftharpoons x^* + RL + ADP$. The active readout proteins $x^*$ can decay spontaneously or be deactivated by an enzyme, such as CheZ in *E. coli*, $x^* \rightleftharpoons x + Pi$. Inside the living cell the system is maintained in a non-equilibrium steady state by keeping the concentrations of ATP, adenosine diphosphate (ADP), and inorganic phosphate (Pi) constant. We absorb their concentrations and the activities of the kinase and, if applicable, phosphatase in the (de)phosphorylation rates, coarse-graining the (de)modification reactions into instantaneous second-order reactions: $x + RL \underset{k_{-f}}{\overset{k_f}{\rightleftharpoons}} x^* + RL$, $x^* \underset{k_{-r}}{\overset{k_r}{\rightleftharpoons}} x$. This system has a relaxation time $\tau_r = 1/[(k_f + k_{-f})\overline{RL} + k_r + k_{-r}]$ (*Govern and Ten Wolde, 2014a*), which describes how fast fluctuations in $x^*$ relax. It determines how long $x^*$ can carry information on the ligand-binding state of the receptor; $\tau_r$ thus sets the integration time of the receptor state.

### Theory: inferring concentration from receptor occupancy

The central idea of our theory is illustrated in *Figure 2a*: the cell employs the push–pull network to estimate the average receptor occupancy $p_{\tau_r}$ over the past integration time $\tau_r$. It then uses this estimate $\hat{p}_{\tau_r}$ to infer the current concentration $L$ via the dynamic input–output relation $p_{\tau_r}(L)$, which provides a one-to-one mapping between $p_{\tau_r}$ and $L$.

#### Dynamic input–output relation

The mapping $p_{\tau_r}(L)$ is the *dynamic input–output relation*. It gives the average receptor occupancy over the past integration time $\tau_r$, *given that the* current *value of the input signal is $L = L(t)$* (see *Figure 2a*). Here, the average is not only over the noise in receptor–ligand binding and readout activation (*Figure 2b*) but also over the subensemble of past input trajectories that each end at the same current concentration $L$ (*Figure 2c*; *Tostevin and ten Wolde, 2010*; *Hilfinger and Paulsson, 2011*; *Bowsher et al., 2013*). In contrast to the conventional static input–output relation $p(L_s)$, which gives the average receptor occupancy $p$ for a steady-state ligand concentration $L_s$ that does not vary in time, the dynamic input–output relation takes into account the dynamics of the input and the finite response time of the system. It depends on all timescales in the problem: the timescale of the input, $\tau_L$, the receptor–ligand correlation time $\tau_c$, and the integration time $\tau_r$. Only when $\tau_L \gg \tau_c, \tau_r$ does the dynamic input–output $p_{\tau_r}(L)$ become equal to the static input–output relation $p(L_s)$.

#### Sensing error

Linearizing the dynamic input–output relation $p_{\tau_r}(L)$ around the mean ligand concentration $\overline{L}$ (see *Figure 2a*) and using the rules of error propagation, the expected error in the concentration estimate is

$$(\delta \hat{L})^2 = \frac{\sigma^2_{\hat{p}_{\tau_r}}}{\tilde{g}^2_{L \to p_{\tau_r}}}. \tag{1}$$

Here, $\sigma^2_{\hat{p}_{\tau_r}}$ is the variance in the estimate $\hat{p}_{\tau_r}$ of the average receptor occupancy over the past $\tau_r$, *given* that the current input signal is $L$ (see *Figure 2a*). The quantity $\tilde{g}_{L \to p_{\tau_r}}$ is the *dynamic* gain, which is the slope of the dynamic input–output relation $p_{\tau_r}(L)$; it determines how much an error in the estimate of $p_{\tau_r}$ propagates to that in $L$. *Equation 1* generalizes the expression for the error in sensing static concentrations (*Berg and Purcell, 1977*; *Bialek and Setayeshgar, 2005*; *Wang et al., 2007*; *Mehta and Schwab, 2012*; *Kaizu et al., 2014*; *Govern and Ten Wolde, 2014a*; *Ten Wolde et al., 2016*) to that of time-varying concentrations.

## Signal-to-noise ratio

Together with the distribution of input states, the sensing error $(\delta \hat{L})^2$ determines how many distinct signal values the cell can resolve. The latter is quantified by the signal-to-noise ratio (SNR), which is defined as

$$\mathrm{SNR} \equiv \frac{\sigma^2_L}{(\delta \hat{L})^2}. \tag{2}$$

Here, $\sigma^2_L$ is the variance of the ligand concentration $L(t)$; because the system is stationary and time invariant, we can omit the argument in $L(t)$ and write $L = L(t)$. The variance $\sigma^2_L$ is a measure for the total number of input states, such that the SNR gives the number of distinct ligand concentrations the cell can measure. Using *Equation 1*, it is given by

$$\mathrm{SNR} = \frac{\tilde{g}^2_{L \to p_{\tau_r}}}{\sigma^2_{\hat{p}_{\tau_r}}} \sigma^2_L. \tag{3}$$

The SNR also yields the mutual information $I(x^*; L) = 1/2 \ln(1 + \mathrm{SNR})$ between the input $L$ and output $x^*$ (*Tostevin and ten Wolde, 2010*).

## Readout system samples receptor state

Receptor time averaging is typically conceived as a scheme in which the receptor state is averaged via the mathematical operation of an integral: $p_{\tau_r} = 1/\tau_r \int_0^{\tau_r} p(t') dt'$. Yet, readout proteins are discrete components that interact with the receptor in a discrete and stochastic fashion. To derive the dynamic gain $\tilde{g}_{L \to p_{\tau_r}}$ and error in estimating $p_{\tau_r}$, $\sigma^2_{\hat{p}_{\tau_r}}$ (*Equation 3*), we therefore view the push–pull network as a device that discretely samples the receptor state (see *Figure 2b*; *Govern and Ten Wolde, 2014a*). The principle is that cells employ the activation reaction $x + RL \to x^* + RL$ to store the state of the receptor in stable chemical modification states of the readout molecules. Readout molecules that collide with a ligand-bound receptor are modified, while those that collide with an unbound receptor are not (*Figure 2b*). The readout molecules serve as samples of the receptor at the time they were created, and collectively they encode the history of the receptor: the fraction of samples that correspond to ligand-bound receptors is the cell's estimate for $p_{\tau_r}$. Indeed, this is the discrete and stochastic implementation of the mechanism of time integration. The effective number of independent samples depends not only on the creation of samples, $x + RL \to x^* + RL$, but also on their decay and accuracy. Samples decay via the deactivation reaction $x^* \to x$, which means that they only provide information on the receptor occupancy over the past $\tau_r$. In addition, both the activation and the deactivation reaction can happen in their microscopic reverse direction, which corrupts the coding, that is, the mapping between the ligand-binding states of the receptor proteins and the activation states of the readout molecules. Energy is needed to break time reversibility and protect the coding. Furthermore, for time-varying signals, we also need to recognize that the samples correspond to the ligand concentration over the past integration time $\tau_r$, which will in general differ from the current concentration $L$ that the cell aims to estimate (see *Figure 2c*). While a finite $\tau_r$ is necessary for time integration, it will, as we show below, also lead to a systematic error in the estimate of the concentration that the cell cannot reduce by taking more receptor samples.

This analysis reveals that the dynamic gain is (see Appendix 1)

$$\tilde{g}_{L \to p_{\tau_r}} = g_{L \to p}\left(1 + \frac{\tau_c}{\tau_L}\right)^{-1}\left(1 + \frac{\tau_r}{\tau_L}\right)^{-1}. \tag{4}$$

Only when $\tau_L \gg \tau_r, \tau_c$ is the average ligand concentration over the ensemble of trajectories ending at $\delta L(t)$ equal to the current concentration $\delta L(t)$ (**Figure 2c**) and does $\tilde{g}_{L \to p_{\tau_r}}$ become equal to its maximal value, the static gain $g_{L \to p} = p(1-p)/\overline{L}$, where $p$ is the average receptor occupancy averaged over all values of $\delta L(t)$. The analysis also reveals that the error in $p_{\tau_r}$ can be written as (see Appendix 1, **Equation 29**)

$$\sigma^2_{\hat{p}_{\tau_r}} = \sigma^{2,\,\text{samp}}_{\hat{p}_{\tau_r}} + \sigma^{2,\,\text{dyn}}_{\hat{p}_{\tau_r}}, \tag{5}$$

where $\sigma^{2,\,\text{samp}}_{\hat{p}_{\tau_r}}$ is a statistical error due to the stochastic sampling of the receptor and $\sigma^{2,\,\text{dyn}}_{\hat{p}_{\tau_r}}$ is a systematic error arising from the dynamics of the input, as elucidated in **Figure 2b, c**.

## Central result

To know how the error $\sigma^2_{\hat{p}_{\tau_r}}$ in the estimate of $p_{\tau_r}$ propagates to the error $(\delta\hat{L})^2$ in the estimate of the current ligand concentration, we divide $\sigma^2_{\hat{p}_{\tau_r}}$ by the dynamic gain $\tilde{g}_{L \to p_{\tau_r}}$ given by **Equation 4** (see **Equation 1**). For the full system, the reversible push–pull network, this yields via **Equation 3** the central result of our article, the SNR in terms of the total number of receptor samples, their independence, their accuracy, and the timescale on which they are generated:

$$\text{SNR}^{-1} = \underbrace{\left(1 + \frac{\tau_c}{\tau_L}\right)^2\left(1 + \frac{\tau_r}{\tau_L}\right)^2\left[\frac{(\overline{L}/\sigma_L)^2}{p(1-p)\overline{N}_I} + \frac{(\overline{L}/\sigma_L)^2}{(1-p)^2\overline{N}_{\text{eff}}}\right]}_{\text{sampling error}} + \underbrace{\left(1 + \frac{\tau_c}{\tau_L}\right)\left(1 + \frac{\tau_r}{\tau_L}\right)\left(1 + \frac{\tau_c\tau_r}{\tau_L(\tau_c + \tau_r)}\right) - 1}_{\text{dynamical error}}. \tag{6}$$

This expression shows that the sensing error $\text{SNR}^{-1}$ can be decomposed into two distinct contributions, which each have a clear interpretation: the *sampling error*, arising from the stochasticity in the sampling of the receptor state, and the *dynamical error*, arising from the dynamics of the input.

When the timescale of the ligand fluctuations $\tau_L$ is much longer than the receptor correlation time $\tau_c$ and the integration time $\tau_r$, $\tau_L \gg \tau_r, \tau_c$, the dynamical error reduces to zero and only the sampling error remains. Here, $\overline{N}_{\text{eff}}$ is the total number of effective samples and $\overline{N}_I$ is the number of these that are independent (**Govern and Ten Wolde, 2014a**). For the full system, they are given by

$$\overline{N}_I = \underbrace{\frac{1}{(1 + 2\tau_c/\Delta)}}_{f_I}\underbrace{\overbrace{\frac{(e^{\beta\Delta\mu_1} - 1)(e^{\beta\Delta\mu_2} - 1)}{e^{\beta\Delta\mu} - 1}}^{q}\overbrace{\frac{\dot{n}\tau_r}{p}}^{\overline{N}}}_{\overline{N}_{\text{eff}}}. \tag{7}$$

The quantity $\dot{n} = k_f p R_T \overline{x} - k_{-f} p R_T \overline{x}^*$ is the net flux of $x$ around the cycle of activation and deactivation, with $R_T$ the total number of receptor proteins and $\overline{x}$ and $\overline{x}^*$ the average number of inactive and active readout molecules, respectively. It equals the rate at which $x$ is modified by the ligand-bound receptor; the quantity $\dot{n}/p$ is thus the sampling rate of the receptor, be it ligand bound or not. Multiplied with the relaxation rate $\tau_r$, it yields the total number of receptor samples $\overline{N}$ obtained during $\tau_r$. However, not all these samples are reliable. The effective number of samples is $\overline{N}_{\text{eff}} = q\overline{N}$, where $0 < q < 1$ quantifies the quality of the sample. Here, $\beta = 1/(k_B T)$ is the inverse temperature, $\Delta\mu_1$ and $\Delta\mu_2$ are the free-energy drops over the activation and deactivation reaction, respectively, with $\Delta\mu = \Delta\mu_1 + \Delta\mu_2$ the total drop, determined by the fuel turnover (see **Figure 1b**). If the system is in thermodynamic equilibrium, $\Delta\mu_1 = \Delta\mu_2 = \Delta\mu = 0$, $q \to 0$ and the system cannot sense because the ligand-binding state of the receptor is equally likely to be copied into the correct modification state of the readout as into the incorrect one. In contrast, if the system is strongly driven out of equilibrium and $\Delta\mu_1, \Delta\mu_2 \to \infty$, then, during each receptor–readout interaction, the receptor state is always copied into the correct activation state of the readout; the sample quality parameter $q$ thus approaches unity and $\overline{N}_{\text{eff}} \to \overline{N}$. Yet, even when all samples are reliable, they may contain redundant

information on the receptor state. The factor $f_{\mathrm{I}}$ is the fraction of the $\overline{N}_{\mathrm{eff}}$ samples that are independent. It reaches unity when the receptor sampling interval $\Delta = 2\tau_{\mathrm{r}}/(\overline{N}_{\mathrm{eff}}/R_{\mathrm{T}})$ becomes larger than the receptor correlation time $\tau_{\mathrm{c}}$.

When the number of samples becomes very large, $\overline{N}_{\mathrm{I}}, \overline{N}_{\mathrm{eff}} \to \infty$, the sampling error reduces to zero. However, the sensing error still contains a second contribution, which, following *Bowsher et al., 2013*, we call the dynamical error. This contribution only depends on timescales. It arises from the fact that the samples encode the receptor history and hence the ligand concentration over the past $\tau_{\mathrm{r}}$, which will, in general, deviate from the quantity that the cell aims to predict – the current concentration $L$. This contribution yields a systematic error, which cannot be eliminated by increasing the number of receptor samples, their independence, or their accuracy. It can only be reduced to zero by making the integration time $\tau_{\mathrm{r}}$ much smaller than the ligand timescale $\tau_{\mathrm{L}}$ (assuming $\tau_{\mathrm{c}}$ is typically much smaller than $\tau_{\mathrm{r}}, \tau_{\mathrm{L}}$). Only in this regime will the ligand concentration in the past $\tau_{\mathrm{r}}$ be similar to the current concentration and can the latter be reliably inferred from the receptor occupancy, provided the latter has been estimated accurately by taking enough samples.

Importantly, the dynamics of the input signal not only affects the sensing precision via the dynamical error but also via the sampling error. This effect is contained in the prefactor of the sampling error, $(1 + \tau_{\mathrm{c}}/\tau_{\mathrm{L}})^2(1 + \tau_{\mathrm{r}}/\tau_{\mathrm{L}})^2$, which has its origin in the dynamic gain $\tilde{g}_{L \to p_{\tau_{\mathrm{r}}}}$ (*Equation 4*). It determines how the sampling error $\sigma_{\hat{p}_{\tau_r}}^{2,\mathrm{samp}}$ in the estimate of $p_{\tau_{\mathrm{r}}}$ propagates to the error in the estimate of $L$ (see *Equation 3*). Only when $\tau_{\mathrm{c}}, \tau_{\mathrm{r}} \ll \tau_{\mathrm{L}}$ can the readout system closely track the input signal and does $\tilde{g}_{L \to p_{\tau_{\mathrm{r}}}}$ reach its maximal value, the static gain $g_{L \to p}$, thus minimizing the error propagation from $p_{\tau_{\mathrm{r}}}$ to $L$.

## Fundamental resources

We can use *Equation 6* to identify the *fundamental* resources for cell sensing (*Govern and Ten Wolde, 2014a*) and derive Pareto fronts that quantify the trade-offs between the maximal sensing precision and these resources. A fundamental resource is a (collective) variable $Q_i$ that, when fixed to a constant, puts a non-zero lower bound on $\mathrm{SNR}^{-1}$, no matter how the other variables are varied. It is thus mathematically defined as $\mathrm{MIN}_{Q_i=\mathrm{const}}(\mathrm{SNR}^{-1}) = f(\mathrm{const}) > 0$. To find these collective variables, we numerically or analytically minimized $\mathrm{SNR}^{-1}$, constraining (combinations of) variables yet optimizing over the other variables. This reveals that the SNR is bounded by (see Appendix 2)

$$\mathrm{SNR}^{-1} \geq \left(1 + \frac{\tau_{\mathrm{r}}}{\tau_{\mathrm{L}}}\right)^2 \frac{4(\overline{L}/\sigma_L)^2}{h} + \frac{\tau_{\mathrm{r}}}{\tau_{\mathrm{L}}}, \tag{8}$$

where

$$h \equiv \mathrm{MIN}(R_{\mathrm{T}}\tau_{\mathrm{r}}/\tau_{\mathrm{c}}, X_{\mathrm{T}}, \beta\dot{w}\tau_{\mathrm{r}}). \tag{9}$$

*Equations 8 and 9* show that the fundamental resources are the number of receptors $R_{\mathrm{T}}$, the integration time $\tau_{\mathrm{r}}$, the number of readouts $X_{\mathrm{T}}$, and the power $\dot{w} = \dot{n}\Delta\mu$.

*Figure 3a, b* illustrates that $R_{\mathrm{T}}, \tau_{\mathrm{r}}, X_{\mathrm{T}}, \dot{w}$ are indeed fundamental: the sensing precision is bounded by the limiting resource and cannot be enhanced by increasing another resource. Panel (a) shows that when $X_{\mathrm{T}}$ is small, the maximum mutual information $I_{\mathrm{max}}$ cannot be increased by raising $R_{\mathrm{T}}$: no matter how many receptors the system has, the sensing precision is limited by the pool of readout molecules and only increasing this pool can raise $I_{\mathrm{max}}$. Yet, when $X_{\mathrm{T}}$ is large, $I_{\mathrm{max}}$ becomes independent of $X_{\mathrm{T}}$. In this regime, the number of receptors $R_{\mathrm{T}}$ limits the number of independent concentration measurements and only increasing $R_{\mathrm{T}}$ can raise $I_{\mathrm{max}}$. Similarly, panel (b) shows that when the power $\dot{w}$ is limiting, $I_{\mathrm{max}}$ cannot be increased by $R_{\mathrm{T}}$ but only by increasing $\dot{w}$. Clearly, the resources receptors, readout molecules, and energy cannot compensate each other: the sensing precision is bounded by the limiting resource.

Importantly, while for sensing static concentrations the products $R_{\mathrm{T}}\tau_{\mathrm{r}}/\tau_{\mathrm{c}}$ and $\dot{w}\tau_{\mathrm{r}}$ are fundamental (*Govern and Ten Wolde, 2014a*), for time-varying signals $R_{\mathrm{T}}, \dot{w}$, and $\tau_{\mathrm{r}}$ separately limit sensing. Consequently, neither receptors $R_{\mathrm{T}}$ nor power $\dot{w}$ can be traded freely against time $\tau_{\mathrm{r}}$ to reach a desired precision, as is possible for static signals. In line with the predictions of signal filtering theories (*Extrapolation, 1950*; *Kolmogorov, 1992*; *Kalman, 1960*), there exists an optimal integration

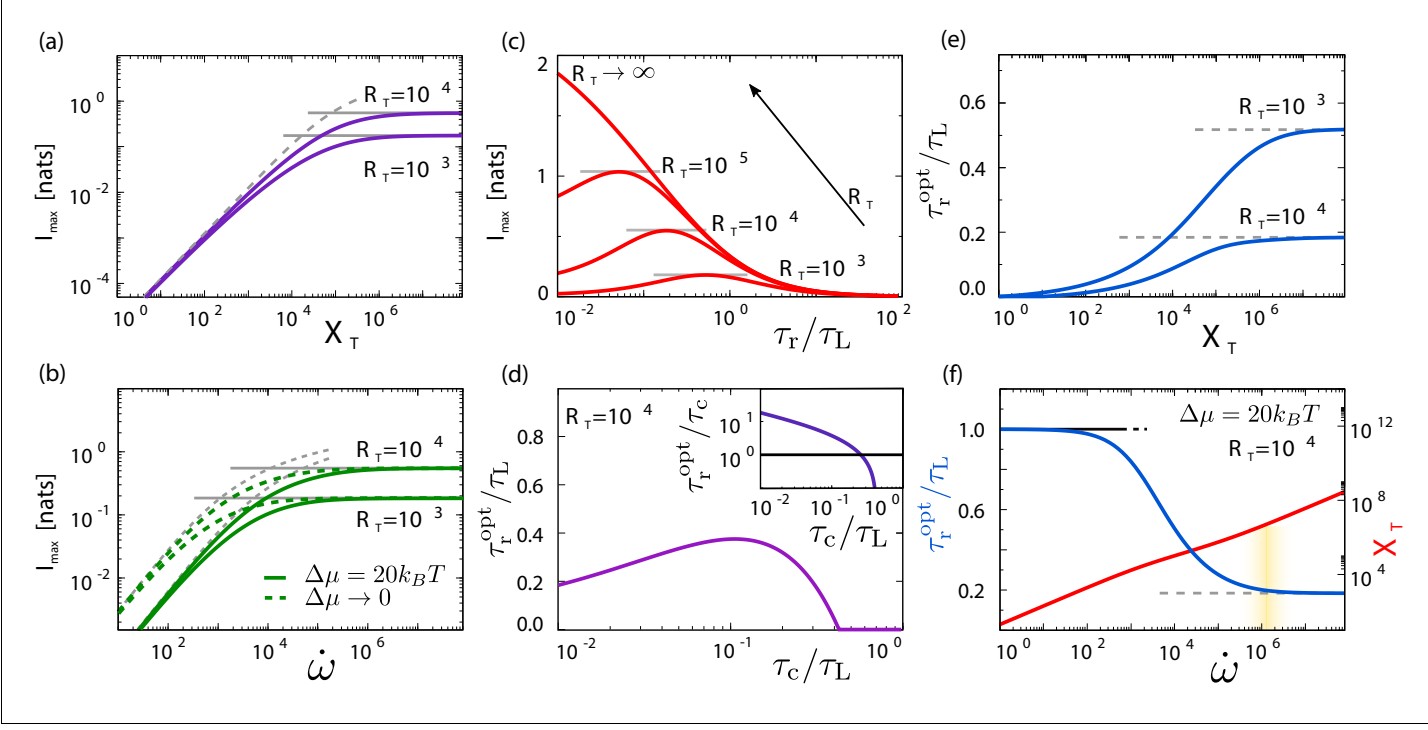

**Figure 3.** Receptors $R_\mathrm{T}$, readout molecules $X_\mathrm{T}$, and $\dot{w}$ fundamentally limit sensing, and there exists an optimal integration time $\tau_\mathrm{r}$ that depends on which of the resources is limiting. (a, b) $R_\mathrm{T}$, $X_\mathrm{T}$, and $\dot{w}$ are fundamental resources, with no trade-offs between them. Plotted is the maximum mutual information $I_\mathrm{max} = 1/2 \ln(1 + \mathrm{SNR}_\mathrm{max})$, obtained by minimizing *Equation 6* over $p$ and $\tau_\mathrm{r}$, for different combinations of (a) $X_\mathrm{T}$ and $R_\mathrm{T}$ in the irreversible limit $q \to 1$ and (b) $\dot{w}$ and $R_\mathrm{T}$ for two different values of $\Delta\mu$. The sensing precision is bounded by the limiting resource, $R_\mathrm{T}$ (solid gray lines, *Equation 8* with $h = R_\mathrm{T}\tau_\mathrm{r}/\tau_\mathrm{c}$), $X_\mathrm{T}$ (dashed gray line, *Equation 8* with $h = X_\mathrm{T}$, panel a), or $\dot{w}$ (dashed gray lines, *Equation 8* with $h = \beta\dot{w}\tau_\mathrm{r}$ or $h = \dot{w}\tau_\mathrm{r}/(\Delta\mu/4)$, panel b). (c) $I_\mathrm{max}$ as a function of $\tau_\mathrm{r}$ for different values of $R_\mathrm{T}$ in the Berg–Purcell limit ($q \to 1$ and $X_\mathrm{T} \to \infty$). There exists an optimal integration time $\tau_\mathrm{r}^\mathrm{opt}$ that maximizes the sensing precision; $\tau_\mathrm{r}^\mathrm{opt}$ decreases with $R_\mathrm{T}$. (d) In this limit, $\tau_\mathrm{r}^\mathrm{opt}$ depends non-monotonically on the receptor–ligand correlation time $\tau_\mathrm{c}$: it first increases with $\tau_\mathrm{c}$ to sustain time-averaging, but then drops when $\tau_\mathrm{r}^\mathrm{opt}/\tau_\mathrm{c}$ becomes of order unity and time-averaging is no longer effective (see inset). (e) $\tau_\mathrm{r}^\mathrm{opt}$ as a function of $X_\mathrm{T}$ for different values of $R_\mathrm{T}$. When $X_\mathrm{T} < R_\mathrm{T}$, time averaging is not possible and the optimal system is an instantaneous responder, $\tau_\mathrm{r}^\mathrm{opt} \to 0$; when $X_\mathrm{T} \gg R_\mathrm{T}$, the system reaches the Berg–Purcell regime in which $I_\mathrm{max}$ is limited by $R_\mathrm{T}$ rather than $X_\mathrm{T}$ (see panel a). (f) $\tau_\mathrm{r}^\mathrm{opt}$ and $X_\mathrm{T}$ as a function of $\dot{w}$. When the power $\dot{w} \sim X_\mathrm{T}/\tau_\mathrm{r}$ is limiting, the sampling error dominates and $\tau_\mathrm{r}^\mathrm{opt}$ equals $\tau_\mathrm{L}$ to maximize $X_\mathrm{T}$, minimizing the sampling error; $\tau_\mathrm{r}^\mathrm{opt}$ then decreases to trade part of the decrease in the sampling error for a reduction in the dynamical error such that both decrease; when the sampling interval $\Delta \sim \tau_\mathrm{r} R_\mathrm{T}/X_\mathrm{T}$ becomes comparable to $\tau_\mathrm{c}$, in the region marked by the yellow bar, the sampling error is no longer limited by $X_\mathrm{T}$, such that $\tau_\mathrm{r}$ now limits both sources of error; the two sources can therefore no longer be decreased simultaneously by increasing $\dot{w} \sim X_\mathrm{T}/\tau_\mathrm{r}$; the system has entered the Berg–Purcell regime, where $\tau_\mathrm{r}^\mathrm{opt}$ is determined by $R_\mathrm{T}$ rather than $\dot{w}$ (see panel b). Parameter values unless specified: $\tau_\mathrm{c}/\tau_\mathrm{L} = 10^{-2}$; $\sigma_L/\overline{L}_T = 10^{-2}$.

time $\tau_\mathrm{r}^\mathrm{opt}$ that maximizes the sensing precision (*Andrews et al., 2006*; *Hinczewski and Thirumalai, 2014*; *Becker et al., 2015*; *Monti et al., 2018b*; *Mora and Nemenman, 2019*). Interestingly, its value depends on which of the resources $R_\mathrm{T}$, $X_\mathrm{T}$, and $\dot{w}$ is limiting (*Figure 3c–f*). We now discuss these three regimes in turn.

## Receptors

*Berg and Purcell, 1977* pointed out that cells can reduce the sensing error by either increasing the number of receptors or taking more measurements per receptor via the mechanism of time integration. However, *Equation 8* reveals that for sensing time-varying signals time integration can never eliminate the sensing error completely, as predicted also by filtering theories (*Extrapolation, 1950*; *Kolmogorov, 1992*; *Kalman, 1960*). *Equation 8* shows that in the *Berg–Purcell* regime, where receptors and their integration time are limiting and $h = R_\mathrm{T}\tau_\mathrm{r}/\tau_\mathrm{c}$, the sensing precision does not depend on $R_\mathrm{T}\tau_\mathrm{r}/\tau_\mathrm{c}$, as for static signals (*Govern and Ten Wolde, 2014a*), but on $R_\mathrm{T}$ and $\tau_\mathrm{r}$ separately, such that an optimal integration time $\tau_\mathrm{r}^\mathrm{opt}$ emerges that maximizes the sensing precision (see *Figure 3c*). Increasing $\tau_\mathrm{r}$ improves the mechanism of time integration by increasing the number of

independent samples per receptor, $\tau_r/\tau_c$, thus reducing the sampling error (*Equation 6*). However, increasing $\tau_r$ raises the dynamical error. Moreover, it lowers the dynamical gain $\tilde{g}_{L \to p_{\tau_r}}$, which increases the propagation of the error in the estimate of the receptor occupancy to that of the ligand concentration. The optimal integration time $\tau_r^{opt}$ arises as a trade-off between these three factors.

*Figure 3c* also shows that the optimal integration time $\tau_r^{opt}$ decreases with the number of receptors $R_T$. The total number of independent concentration measurements is the number of independent measurements per receptor, $\tau_r/\tau_c$, times the number $R_T$ of receptors, $\overline{N}_I = R_T \tau_r/\tau_c$. As $R_T$ increases, less measurements $\tau_r/\tau_c$ per receptor have to be taken to remove the receptor–ligand-binding noise, explaining why $\tau_r^{opt}$ decreases as $R_T$ increases – time integration becomes less important.

Interestingly, $\tau_r^{opt}$ depends non-monotonically on the receptor–ligand correlation time $\tau_c$ (*Figure 3d*). When $\tau_c$ increases at fixed $\tau_r$, the receptor samples become more correlated. To keep the mechanism of time integration effective, $\tau_r$ must increase with $\tau_c$. However, to avoid too strong signal distortion the cell compromises on time integration by decreasing the *ratio* $\tau_r/\tau_c$ (see inset). When $\tau_r$ becomes too large, the benefit of time integration no longer pays off the cost of signal distortion. Now not only the ratio $\tau_r/\tau_c$ decreases but also $\tau_r$ itself. The sensing system switches to a different strategy: it no longer employs time integration but becomes an instantaneous sensor.

## Readout molecules

To implement time integration, the cell needs to store the receptor states in the readout molecules. When the number of readout molecules $X_T$ is limiting, the sensing precision is given by *Equation 8* with $h = X_T$. This bound is saturated when $\tau_r \to 0$. This is in marked contrast to the non-zero optimal integration $\tau_r^{opt}$ in the Berg–Purcell regime (see *Figure 3c*).

To elucidate the non-trivial behavior of $\tau_r^{opt}$, *Figure 3e* shows $\tau_r^{opt}$ as a function of $X_T$. When $X_T$ is smaller than $R_T$, the average number of samples per receptor is less than unity. In this regime, the system cannot time integrate the receptor, and to minimize signal distortion $\tau_r^{opt} \approx 0$. Yet, when $X_T$ is increased, the likelihood that two or more readout molecules provide a sample of the same receptor molecule rises, and time averaging becomes possible. Yet to obtain receptor samples that are independent, the integration time $\tau_r$ must be increased to make the sampling interval $\Delta \sim \tau_r R_T/X_T$ larger than the receptor correlation time $\tau_c$. As $X_T$ and hence the total number of samples $\overline{N}$ are increased further, the number of samples that are independent, $\overline{N}_I$, only continues to rise when $\tau_r$ increases with $X_T$ further. However, while this reduces the sampling error, it also increases the dynamical error. When the decrease in the sampling error no longer outweighs the increase in the dynamical error, $\tau_r^{opt}$ and the mutual information no longer change with $X_T$ (see *Figure 3a*). The system has entered the Berg–Purcell regime in which $\tau_r^{opt}$ and the mutual information are given by the optimization of *Equation 8* with $h = R_T \tau_r/\tau_c$ (gray dashed line). In this regime, increasing $X_T$ merely adds redundant samples: the number of independent samples remains $\overline{N}_I = R_T \tau_r^{opt}/\tau_c$.

## Power

Time integration relies on copying the ligand-binding state of the receptor into the chemical modification states of the readout molecules (*Mehta and Schwab, 2012*; *Govern and Ten Wolde, 2014a*). This copy process correlates the state of the receptor with that of the readout, which requires work input (*Ouldridge et al., 2017*).

The free-energy $\Delta\mu$ provided by the fuel turnover drives the readout around the cycle of modification and demodification (*Figure 1*). The rate at which the fuel molecules do work is the power $\dot{w} = \dot{n}\Delta\mu$, and the total work performed during the integration time $\tau_r$ is $w \equiv \dot{w}\tau_r$. This work is spent on taking samples of receptor molecules that are bound to ligand because only they can modify the readout. The total number of effective samples of ligand-bound receptors during $\tau_r$ is $p\overline{N}_{eff}$ (*Equation 7*), which means that the work per effective sample of a ligand-bound receptor is $w/(p\overline{N}_{eff}) = \Delta\mu/q$ (*Govern and Ten Wolde, 2014a*).

To understand how energy limits the sensing precision, we can distinguish between two limiting regimes (*Govern and Ten Wolde, 2014a*). When $\Delta\mu > 4k_BT$, the quality parameter $q \to 1$ (*Equation 7*) and the work per sample of a ligand-bound receptor is $w/(p\overline{N}_{eff}) = \Delta\mu$ (*Govern and Ten Wolde,*

*2014a*). In this irreversible regime, the SNR bound is given by *Equation 8* with $h = \dot{w}\tau_r/(\Delta\mu/4)$. The power limits the sensing accuracy not because it limits the reliability of each sample but because it limits the rate $\dot{n} = \dot{w}/\Delta\mu$ at which the receptor is sampled.

When $\Delta\mu < 4k_BT$, the system enters the quasi-equilibrium regime in which the quality parameter $q \to \beta\Delta\mu/4$ (see *Equation 7*, noting that in the optimal system $\Delta\mu_1 = \Delta\mu_2 = \Delta\mu/2$). The sensing bound is now given by *Equation 8* with $h = \beta\dot{w}\tau_r$, which is larger than $h = \dot{w}\tau_r/(\Delta\mu/4)$ in the irreversible regime (where $\Delta\mu > 4k_BT$). The quasi-equilibrium regime minimizes the sensing error for a given power constraint (*Figure 3b*) because this regime maximizes the number of effective measurements per work input $p\overline{N}_{\text{eff}}/w = q/\Delta\mu = \beta/4$ (*Govern and Ten Wolde, 2014a*).

While the sensing precision for a given power and time constraint is higher in the quasi-reversible regime, more readout molecules are required to store the concentration measurements in this regime. Noting that the flux $\dot{n} = f(1-f)X_Tq/\tau_r = \dot{w}/\Delta\mu$, it follows that in the irreversible regime ($q \to 1$) the number of readout molecules consuming energy at a rate $\dot{w}$ is

$$X_T^{\text{irr}} = \frac{\dot{w}\tau_r}{\Delta\mu f(1-f)}, \tag{10}$$

while in the quasi-equilibrium regime ($q \to \Delta\mu/4$) it is

$$X_T^{\text{qeq}} = \frac{\dot{w}\tau_r 4k_BT}{\Delta\mu^2 f(1-f)}. \tag{11}$$

Since in the quasi-equilibrium regime $\Delta\mu < 4k_BT$, $X_T^{\text{qeq}} > X_T^{\text{irr}}$.

*Equation 8* shows that the sensing precision is fundamentally bounded not by the work $w = \dot{w}\tau_r$, as observed for static signals (*Govern and Ten Wolde, 2014a*), but rather by the power $\dot{w}$ and the integration time $\tau_r$ separately such that an optimal integration time $\tau_r^{\text{opt}}$ emerges. *Figure 3f* shows how $\tau_r^{\text{opt}}$ depends on $\dot{w}$. Since the system cannot sense without any readout molecules, in the low-power regime the system maximizes $X_T$ subject to the power constraint $\dot{w} \sim X_T/\tau_r$ (see *Equations 10 and 11*) by making $\tau_r$ as large as possible, which is the signal correlation time $\tau_L$ – increasing $\tau_r^{\text{opt}}$ further would average out the signal itself. As $\dot{w}$ is increased, $X_T$ rises and the sampling error decreases. When the sampling error becomes comparable to the dynamical error (*Equation 6*), the system starts to trade a further reduction in the sampling error for a reduction in the dynamical error by decreasing $\tau_r^{\text{opt}}$. The sampling error and dynamical error are now reduced simultaneously by increasing $X_T$ and decreasing $\tau_r^{\text{opt}}$. This continues until the sampling interval $\Delta \sim R_T\tau_r/X_T$ becomes comparable to the receptor correlation time $\tau_c$, as marked by the yellow bar. Beyond this point, $\Delta < \tau_c$ and the sampling error is no longer limited by $X_T$ but rather by $\tau_r$ since $\tau_r$ bounds the number of *independent* samples per receptor, $\tau_r/\tau_c$. The system has entered the Berg–Purcell regime, where $\tau_r^{\text{opt}}$ is determined by the trade-off between the dynamical error and the sampling error as set by the maximum number of independent samples, $R_T\tau_r/\tau_c$ (*Figure 3c*).

## Optimal design

In sensing time-varying signals, a trade-off between time averaging and signal tracking is inevitable. Moreover, the optimal integration time depends on which resource is limiting, being zero when $X_T$ is limiting and finite when $R_T$ or $\dot{w}$ is limiting (*Figure 3*). It is therefore not obvious whether these sensing systems still obey the optimal resource allocation principle as observed for systems sensing static concentrations (*Govern and Ten Wolde, 2014a*).

However, *Equation 8* shows that when for a *given* integration time $\tau_r$, $R_T\tau_r/\tau_c = X_T = \beta\dot{w}\tau_r$, the bounds on the sensing precision as set by, respectively, the number of receptors $R_T$, the number of readout molecules $X_T$, and the power $\dot{w}$ are equal. Each of these resources is then equally limiting sensing and no resource is in excess. We thus recover the optimal resource allocation principle:

$$R_T\tau_r/\tau_c \approx X_T \approx \beta\dot{w}\tau_r. \tag{12}$$

Irrespective of whether the concentration fluctuates in time, the number of independent concentration measurements at the receptor level is $R_T\tau_r/\tau_c$, which in an optimally designed system also equals the number of readout molecules $X_T$ and the energy $\beta\dot{w}\tau_r$ that are both necessary and sufficient to store these measurements reliably.

The design principle $X_T \approx \beta \dot{w} \tau_r$ (*Equation 12*) predicts that there exists a driving force $\Delta\mu^{opt}$ that optimizes the trade-off between the number of samples and their accuracy. Noting that $\beta \dot{w} \tau_r = \beta \dot{n} \Delta\mu \tau_r = \beta q f (1-f) X_T \Delta\mu$ reveals that the principle $X_T \approx \beta \dot{w} \tau_r$ (*Equation 12*) specifies $\Delta\mu$ for the optimal system in which $f \to 1/2$ and $\Delta\mu_1 = \Delta\mu_2 = \Delta\mu/2$ via the equation $q(\Delta\mu^{opt}) = 4k_B T / \Delta\mu^{opt}$, where $q(\Delta\mu)$ is defined in *Equation 7*. A numerical inspection shows that to a good approximation the solution of this equation is precisely given by the crossover from the quasi-equilibrium regime to the irreversible one: $\Delta\mu^{opt} \approx 4k_B T$. This can be understood by noting that in the quasi-equilibrium regime $X_T$ can, for a given power and time constraint, be reduced by increasing $\Delta\mu$ (*Equation 11*) *without compromising the sensing precision* (*Equation 8* with $h = \dot{w}\tau_r$); in this regime, increasing $\Delta\mu$ increases the reliability of each sample, and a smaller number of more reliable samples precisely compensates for a larger number of less reliable ones. Yet, when $\Delta\mu$ becomes larger than $4k_B T$, the system enters the irreversible regime. Here, $X_T$ corresponding to a given $\dot{w}$ and $\tau_r$ constraint still decreases with $\Delta\mu$ (*Equation 10*), but the sensing error now increases (*Equation 8* with $h = \dot{w}\tau_r / (\Delta\mu/4)$) because each sample has become (essentially) perfect in this regime – hence, the samples' accuracy cannot (sufficiently) increase further to compensate for the reduction in the sampling rate $\dot{n} \sim X_T / \tau_r$.

*Equation 12* holds for any integration time $\tau_r$, yet it does not specify $\tau_r$. The cell membrane is highly crowded, and many systems employ time integration (*Berg and Purcell, 1977*; *Bialek and Setayeshgar, 2005*; *Govern and Ten Wolde, 2014a*). This suggests that these systems employ time integration and accept the signal distortion that comes with it simply because there is not enough space on the membrane to increase $R_T$. Our theory then allows us to predict the optimal integration time $\tau_r^{opt}$ based on the premise that $R_T$ is limiting. As *Equation 8* reveals, in this limit $\tau_r^{opt}$ does not only depend on $R_T$ but also on $\tau_c$, $\tau_L$, and $\sigma_L/\overline{L}$ : $\tau_r^{opt} = \tau_r^{opt}(R_T, \tau_r, \tau_L, \sigma_L/\overline{L})$. The optimal design of the system is then given by *Equation 12* but with $\tau_r$ given by $\tau_r^{opt} = \tau_r^{opt}(R_T, \tau_c, \tau_L, \sigma_L/\overline{L})$ :

$$R_T \tau_r^{opt} / \tau_c \approx X_T^{opt} \approx \beta \dot{w}^{opt} \tau_r^{opt}. \tag{13}$$

This design principle maximizes for a given number of receptors $R_T$ the sensing precision and minimizes the number of readout molecules $X_T$ and power $\dot{w}$ needed to reach that precision.

## Comparison with experiment

To test our theory, we turn to the chemotaxis system of *E. coli*. This system contains a receptor that forms a complex with the kinase CheA. This complex, which is coarse-grained into $R$ (*Govern and Ten Wolde, 2014a*), can bind the ligand L and activate the intracellular messenger protein CheY ($x$) by phosphorylating it. Deactivation of CheY is catalyzed by CheZ, the effect of which is coarse-grained into the deactivation rate. This push–pull network allows *E. coli* to measure the current concentration, and the relaxation time of this network sets the integration time for the receptor (*Sartori and Tu, 2011*). The system also exhibits adaptation on longer timescales due to receptor methylation and demethylation. The push–pull network and the adaptation system together allow the cell to measure concentration gradients via a temporal derivative, taking the difference between the current concentration and the past concentration as set by the adaptation time (*Segall et al., 1986*). A *lower* bound for the error in the estimate of this difference is given by the error in the estimate of the current concentration, the central quantity of our theory. Here, we ask how accurately *E. coli* can estimate the latter and whether the sensing precision is sufficient to determine whether during a run the concentration has changed.

Our theory predicts that if the number of receptors is limiting then the optimal integration time $\tau_r^{opt}(R_T, \tau_c, \tau_L, \sigma_L/\overline{L})$ is given by minimizing *Equation 8* with $h = R_T \tau_r / \tau_c$. The number of receptor–CheA complexes depends on the growth rate and varies between $R_T \approx 10^3$ and $R_T \approx 10^4$ (*Li and Hazelbauer, 2004*). The receptor correlation time for the binding of aspartate to the Tar receptor can be estimated from the measured dissociation constant (*Vaknin and Berg, 2007*) and the association rate (*Danielson et al., 1994*), $\tau_c \approx 10 \text{ms}$ (*Govern and Ten Wolde, 2014a*). The timescale $\tau_L$ of the input fluctuations is set by the typical run time, which is on the order of a few seconds, $\tau_L \approx 1\text{s}$ (*Berg and Brown, 1972*; *Taute et al., 2015*).

This leaves one parameter to be determined, $(\sigma_L/\overline{L})^2$. This is set by the spatial ligand–concentration profile and the typical length of a run. We have a good estimate of the latter. In shallow gradients, it is on the order of $l \approx 50\mu\text{m}$ (*Berg and Brown, 1972*; *Taute et al., 2015*; *Jiang et al., 2010*;

*Flores et al., 2012*); specifically, Figure 4 of *Taute et al., 2015* shows that the typical run times are 1–2 s while the typical run speeds are $20 - 60 \mu \mathrm{ms}^{-1}$, yielding a run length on the order of indeed 50 μm. We do not know the spatial concentration profiles that *E. coli* has experienced during its evolution. We can however get a sense of the scale by considering an exponential ligand–concentration gradient. For a profile $\overline{L}(x) = L_0 e^{x/x_0}$ with length scale $x_0$, the relative change in the signal over the length of a run is $\sigma_L/\overline{L} \simeq (d\overline{L}/dx)l/\overline{L} = l/x_0$. We consider the range $\sigma_L/\overline{L} \approx l/x_0 < 1$, where $\sigma_L/\overline{L} < 0.1$ corresponds to shallow gradients with $x_0 \gtrsim 500 \mu \mathrm{m}$ in which cells move with a constant drift velocity (*Shimizu et al., 2010*; *Flores et al., 2012*).

*Figure 4a* shows that as the gradient becomes steeper and $\sigma_L/\overline{L} \approx l/x_0$ increases the optimal integration time $\tau_\mathrm{r}^\mathrm{opt}$ decreases. This can be understood by noting that the relative importance of the dynamical error compared to the sampling error scales with $(\sigma_L/\overline{L})^2$ (*Equation 6*). Shallow ingredients thus allow for a larger integration time while steep gradients necessitate a shorter one.

Experiments indicate that the relaxation rate of CheY is $\tau_\mathrm{r}^{-1} \approx 2 \mathrm{s}^{-1}$ for the attractant response and $\approx 20 \mathrm{s}^{-1}$ for the repellent response (*Sourjik and Berg, 2002*), such that the integration time $\tau_\mathrm{r} \approx 50 - 500 \mathrm{ms}$ (*Sourjik and Berg, 2002*; *Govern and Ten Wolde, 2014a*). *Figure 4a* shows that this integration time is optimal for detecting shallow gradients. Our theory thus predicts that the *E. coli* chemotaxis system has been optimized for sensing shallow gradients.

To navigate, the cells must be able to resolve the signal change over a run. During a run of duration $\tau_\mathrm{L}$, the system performs $\tau_\mathrm{L}/\tau_\mathrm{r}$ independent concentration measurements. The effective error for these measurements is the instantaneous sensing error $(\delta\hat{L})^2$ divided by the number of independent measurements $\tau_\mathrm{L}/\tau_\mathrm{r} : (\delta\hat{L})^2/(\tau_\mathrm{L}/\tau_\mathrm{r})$. Hence, the SNR for these concentration measurements is $\mathrm{SNR}_{\tau_\mathrm{L}} \equiv (\sigma_L/\delta\hat{L})^2 \tau_\mathrm{L}/\tau_\mathrm{r}$.

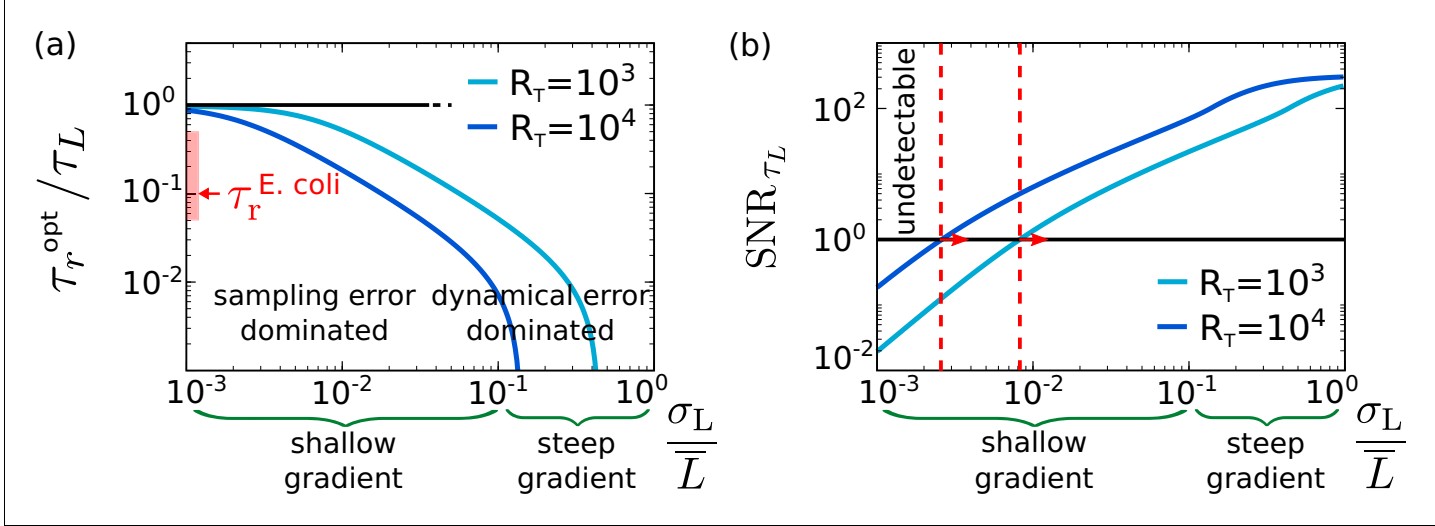

**Figure 4.** The optimal integration time for the chemotaxis system of *E. coli*. (a) The optimal integration time $\tau_\mathrm{r}^\mathrm{opt}$, obtained by numerically optimizing *Equation 8* with $h = R_\mathrm{T}\tau_\mathrm{r}/\tau_\mathrm{c}$, as a function of the relative strength of the input noise, $\sigma_L/\overline{L}$, for two different copy numbers $R_\mathrm{T}$ of the receptor–CheA complexes; for an exponential gradient with length scale $x_0$, the relative noise strength $\sigma_L/\overline{L} \simeq l/x_0$, where $l \approx 50 \mu \mathrm{m}$ is the run length of *E. coli*. It is seen that $\tau_\mathrm{r}^\mathrm{opt}$ increases as $\sigma_L/\overline{L}$ decreases because the relative importance of the sampling error compared to the dynamical error increases. The figure also shows that $\tau_\mathrm{r}^\mathrm{opt}$ decreases as $R_\mathrm{T}$ is increased because that allows for more instantaneous measurements (see also *Figure 3*). The red bar indicates the range of the estimated integration time of *E. coli*, $50 \mathrm{ms} < \tau_\mathrm{r} < 500 \mathrm{ms}$, based on its attractant and repellent response, respectively (*Sourjik and Berg, 2002*), divided by the input timescale $\tau_\mathrm{L} \approx 1 \mathrm{s}$ based on its typical run time of about a second (*Berg and Brown, 1972*; *Taute et al., 2015*). The panel indicates that *E. coli* has been optimized to detect shallow concentration gradients. (b) The signal-to-noise ratio $\mathrm{SNR}_{\tau_\mathrm{L}} = (\sigma_L/\delta\hat{L})^2 \tau_\mathrm{L}/\tau_\mathrm{r}$, with $(\sigma_L/\delta\hat{L})^2 = \mathrm{SNR}$ given by *Equation 6*, as a function of $\sigma_L/\overline{L} \simeq l/x_0$. To be able to detect the gradient, the $\mathrm{SNR}_{\tau_\mathrm{L}}$ must exceed unity. The panel shows that the shallowest gradient that *E. coli* can detect (marked with dashed red line) has, for $R_\mathrm{T} = 10^4$, a length scale of $x_0 \approx 25000 \mu \mathrm{m}$ (corresponding to $\sigma_L/\overline{L} \approx 2 \times 10^{-3}$), which is consistent with experiments based on ramp responses (*Shimizu et al., 2010*). Other parameter: receptor–ligand-binding correlation time $\tau_\mathrm{c} = 10 \mathrm{ms}$ (*Vaknin and Berg, 2007*; *Danielson et al., 1994*).

*Figure 4b* shows that our theory predicts that when $R_\mathrm{T} = 10^3$, the shallowest gradient that cells can resolve, defined by $\mathrm{SNR}_{\tau_\mathrm{L}} = 1$, is $l/x_0 \approx \sigma_L/\overline{L} \approx 1 \times 10^{-2}$, corresponding to $x_0 \approx 7500\mu\mathrm{m}$, while when $R_\mathrm{T} = 10^4$, $l/x_0 \approx 2 \times 10^{-3}$ and $x_0 \approx 25000\mu\mathrm{m}$. The shallowest gradient is thus on the order of $x_0 \approx 10^4\mu\mathrm{m}$. *Shimizu et al., 2010* show that *E. coli* cells are indeed able to sense such very shallow gradients: Figure 2A of *Shimizu et al., 2010* shows that *E. coli* cells can detect exponential up ramps with rate $r = 0.001/\mathrm{s}$; using $r = v_\mathrm{r}/x_0$, where $v_\mathrm{r} \approx 10\mu\mathrm{m/s}$ is the run speed (*Jiang et al., 2010*), this corresponds to $x_0 \approx 10^4\mu\mathrm{m}$. Importantly, the predictions of our theory (*Figure 4*) concern the shallowest gradient that the system with the optimal integration time can resolve. These observations indicate that the optimal integration time is not only sufficient to make navigation in these very shallow gradients possible but also necessary.

*Figure 4* also shows that $\tau_\mathrm{r}^{\mathrm{opt}}$ decreases as the number of receptor–CheA complex, $R_\mathrm{T}$, increases because the latter allows for more instantaneous measurements, reducing the need for time integration (*Figure 3c*). Interestingly, the data of *Li and Hazelbauer, 2004* shows that the copy numbers of the chemotaxis proteins vary with the growth rate. Clearly, it would be of interest to directly measure the response time in different strains under different growth conditions.

## Discussion

Here, we have integrated ideas from *Tostevin and ten Wolde, 2010*; *Hilfinger and Paulsson, 2011*; and *Bowsher et al., 2013* on information transmission via time-varying signals with the sampling framework of *Govern and Ten Wolde, 2014a* to develop a unified theory of cellular sensing. The theory is founded on the concept of the dynamic input–output relation $p_{\tau_\mathrm{r}}(L)$. It allows us to develop the idea that the cell employs the readout system to estimate the average receptor occupancy $p_{\tau_\mathrm{r}}$ over the past integration time $\tau_\mathrm{r}$ and then exploits the mapping $p_{\tau_\mathrm{r}}(L)$ to estimate the current ligand concentration $L$ from $p_{\tau_\mathrm{r}}$. The theory reveals that the error in the estimate of $L$ depends on how accurately the cell samples the receptor state to estimate $p_{\tau_\mathrm{r}}$, and on how much $p_{\tau_\mathrm{r}}$, which is determined by the concentration in the past $\tau_\mathrm{r}$, reflects the current ligand concentration. These two distinct sources of error give rise to the sampling error and dynamical error in *Equation 6*, respectively.

While the system contains no less than 11 parameters, *Equation 6* provides an intuitive expression for the sensing error in terms of collective variables that have a clear interpretation. The dynamical error depends only on the timescales in the problem, most notably $\tau_\mathrm{r}/\tau_\mathrm{L}$. The sampling error depends on how accurately the readout system estimates $p_{\tau_\mathrm{r}}$, which is determined by the number of receptor samples, their independence, and their accuracy; yet it also depends on $\tau_\mathrm{r}/\tau_\mathrm{L}$ via the dynamic gain, which determines how the error in the estimate of $p_{\tau_\mathrm{r}}$ propagates to that of $L$. The trade-off between the sampling error and dynamical error yields an optimal integration time.

Our study reveals that the optimal integration time $\tau_\mathrm{r}^{\mathrm{opt}}$ depends in a non-trivial manner on the design of the system. When the number of readout molecules $X_\mathrm{T}$ is smaller than the number of receptors $R_\mathrm{T}$, time integration is not possible and the optimal system is an instantaneous responder with $\tau_\mathrm{r}^{\mathrm{opt}} \approx 0$. When the power $\dot{w} \sim X_\mathrm{T}/\tau_\mathrm{r}$, rather than $X_\mathrm{T}$, is limiting, $\tau_\mathrm{r}^{\mathrm{opt}}$ is determined by the trade-off between the sampling error and dynamical error. In both scenarios, however, one resource, $X_\mathrm{T}$ or $\dot{w}$, is limiting the sensing precision. In an optimally designed system, all resources are equally limiting so that no resource is wasted. This yields the resource allocation principle (*Equation 12*), first identified in *Govern and Ten Wolde, 2014a*, for sensing static concentrations. The reason it can be generalized to time-varying signals is that the principle concerns the optimal design of the readout system for estimating the receptor occupancy over a given integration time $\tau_\mathrm{r}$, which holds for any type of input: the number of independent concentration measurements at the receptor level is $R_\mathrm{T}\tau_\mathrm{r}/\tau_\mathrm{c}$, irrespective of how the input varies, and in an optimally designed system this also equals the number of readout molecules $X_\mathrm{T}$ and energy $\beta\dot{w}\tau_\mathrm{r}$ to store these measurements reliably. We thus expect that the design principle also holds for systems that sense signals that vary more strongly in time (*Mora and Nemenman, 2019*).

While the allocation principle *Equation 12* holds for any $\tau_\mathrm{r}$, it does not specify the optimal integration time $\tau_\mathrm{r}^{\mathrm{opt}}$. However, our theory predicts that if the number of receptors $R_\mathrm{T}$ is limiting, then there exists a $\tau_\mathrm{r}^{\mathrm{opt}}$ that maximizes the sensing precision for that $R_\mathrm{T}$ (*Equation 8* with $h = R_\mathrm{T}\tau_\mathrm{r}/\tau_\mathrm{c}$). Via the allocation principle *Equation 13*, $R_\mathrm{T}$ and $\tau_\mathrm{r}^{\mathrm{opt}}$ then together determine the minimal number

of readout molecules $X_\mathrm{T}$ and power $\dot{w}$ to reach that precision. The resource allocation principle, together with the optimal integration time, thus completely specifies the optimal design of the sensing system.

Applying our theory to the *E. coli* chemotaxis system shows that this system not only obeys the resource allocation principle (*Govern and Ten Wolde, 2014a*) but also that the predicted optimal integration time to measure shallow gradients is in agreement with that measured experimentally (*Figure 4a*). This is remarkable because there is not a single fit parameter in our theory. Moreover, *Figure 4b* shows that the optimal integration time is not only sufficient to enable the sensing of these shallow gradients but also necessary. This is interesting because the sensing precision could also be increased by increasing the number of receptors, readout molecules, and energy devoted to sensing – but this would be costly. Our results thus demonstrate not only that the chemotaxis system obeys the design principles as revealed by our theory but also that there is a strong selection pressure to design sensing systems optimally, that is, to maximize the sensing precision given the resource constraints.

Our theory is based on a Gaussian model and describes the optimal sensing system that minimizes the mean square error in the estimate of the ligand concentration (see *Equation 1)*. The latter is precisely the performance criterion of Wiener–Kolmogorov (*Extrapolation, 1950*; *Kolmogorov, 1992*) and *Kalman, 1960* filtering theory, which, moreover, become exact for systems that obey Gaussian statistics. In fact, since our system (including the input signal) is stationary, they predict the same optimal filter, which is an exponential filter for signals that are memoryless. The signals studied here belong to this class, and the push–pull network forms an exponential filter (*Hinczewski and Thirumalai, 2014*; *Becker et al., 2015*). This underscores the idea that our theory gives a complete description, in terms of all the required resources, for the optimal design of cellular sensing systems that need to estimate this type of signals. Furthermore, because our model is Gaussian, the goal of minimizing the mean-square error in the estimate of the input signal is equivalent to maximizing the mutual information between the input (the ligand concentration) and the output (the readout $x^*$) (*Becker et al., 2015*).

In recent years, filtering theories and information theory have been applied increasingly to neuronal and cellular systems (*Laughlin, 1981*; *Brenner et al., 2000*; *Fairhall et al., 2001*; *Andrews et al., 2006*; *Ziv et al., 2007*; *Nemenman et al., 2008*; *Cheong et al., 2011*; *Nemenman, 2012*; *Hinczewski and Thirumalai, 2014*; *Becker et al., 2015*; *Husain et al., 2019*; *Tkacik et al., 2008*; *Tkačik and Walczak, 2011*; *Dubuis et al., 2013*; *Monti and Wolde, 2016*; *Monti et al., 2018a*). A key concept in these theories is that optimal sensing systems match the response to the statistics of the input. When the noise is weak, maximizing the entropy of the output distribution becomes paramount, which entails matching the shape of the input–output relation to the shape of the input distribution to generate a flat output distribution (*Laughlin, 1981*; *Tkacik et al., 2008*; *Monti et al., 2018a*). Yet, when the noise is large, the optimal response is also shaped by the requirement to tame the propagation of noise in the input signal (*Andrews et al., 2006*; *Hinczewski and Thirumalai, 2014*; *Becker et al., 2015*; *Monti et al., 2018a*; *Monti et al., 2018b*; *Mora and Nemenman, 2019*) or to lift the signal above the intrinsic noise in the response system (*Tostevin and ten Wolde, 2010*; *Bowsher et al., 2013*). In Appendix 3, we show that estimating the concentration from $p_{\tau_\mathrm{r}}$ is equivalent to that via readout $x^*$. This makes it possible to connect our sampling framework, which is based on $p_{\tau_\mathrm{r}}(L)$, to filtering and information theory, which are based on $x^*(L)$. In particular, we show in this appendix how the optimal integration and dynamic gain can be understood from these ideas on matching the response to the input. We also briefly discuss in Appendix 3 the concepts from information theory that are beyond the scope of the Gaussian model considered here.

Yet, our discrete sampling framework gives a detailed description of how the optimal design of sensing systems depends on the statistics of the input signal in terms of all the required cellular resources: protein copies, time, and energy. In an optimal system, each receptor is sampled once every receptor–ligand correlation time $\tau_\mathrm{c}$, $\Delta \approx \tau_\mathrm{c}$, and the number of samples per receptor is $\tau_\mathrm{r}^{\mathrm{opt}}/\Delta \approx \tau_\mathrm{r}^{\mathrm{opt}}/\tau_\mathrm{c}$. The optimal integration time $\tau_\mathrm{r}^{\mathrm{opt}}$ for a given $R_\mathrm{T}$ is determined by the trade-off between the age of the samples and the number required for averaging the receptor state. When the input varies more rapidly, the samples need to be refreshed more regularly: to keep the dynamical error and the dynamic gain constant, $\tau_\mathrm{r}^{\mathrm{opt}}$ must decrease linearly with $\tau_\mathrm{L}$ (see *Equation 6)*. Yet,

only decreasing $\tau_{\mathrm{r}}^{\mathrm{opt}}$ would inevitably increase the sampling error $\sigma_{\hat{p}_{\tau_r}}^{2,\mathrm{samp}}$ in estimating the receptor occupancy because the sampling interval $\Delta \sim R_{\mathrm{T}} \tau_{\mathrm{r}}^{\mathrm{opt}}/X_{\mathrm{T}}^{\mathrm{opt}}$ would become smaller than $\tau_{\mathrm{c}}$, creating redundant samples. To keep the sensing precision constant, the number of receptors $R_{\mathrm{T}}$ needs to be raised with $\tau_{\mathrm{L}}^{-1}$, such that the sampling interval $\Delta \sim R_{\mathrm{T}} \tau_{\mathrm{r}}^{\mathrm{opt}}/X_{\mathrm{T}}^{\mathrm{opt}}$ remains of order $\tau_{\mathrm{c}}$ and the decrease in the number of samples per receptor, $\tau_{\mathrm{r}}^{\mathrm{opt}}/\tau_{\mathrm{c}}$, is precisely compensated for by the increase in $R_{\mathrm{T}}$. The total number of independent concentration measurements, $R_{\mathrm{T}} \tau_{\mathrm{r}}^{\mathrm{opt}}/\tau_{\mathrm{c}}$, and hence the number of readout molecules $X_{\mathrm{T}}^{\mathrm{opt}}$ to store these, does indeed not change. In contrast, the required power $\beta \dot{w}^{\mathrm{opt}} \approx R_{\mathrm{T}}/\tau_{\mathrm{c}}$ rises (*Equation 12*): each receptor molecule is sampled each $\tau_{\mathrm{c}}$ at $\Delta\mu^{\mathrm{opt}} \approx 4 k_{\mathrm{B}}T$, and the increase in $R_{\mathrm{T}}$ raises the sampling rate $\dot{n} = \dot{w}^{\mathrm{opt}}/\Delta\mu^{\mathrm{opt}} \sim X_{\mathrm{T}}^{\mathrm{opt}}/\tau_{\mathrm{r}}^{\mathrm{opt}}$. Our theory thus predicts that when the input varies more rapidly the number of receptors and the power must rise to maintain a required sensing precision, while the number of readout molecules does not.

The fitness benefit of a sensing system does not only depend on the sensing precision but also on the energetic cost of maintaining and running the system. In principle, the cell can reduce the sensing error arbitrarily by increasing $R_{\mathrm{T}}$ and decreasing $\tau_{\mathrm{r}}$. Our resource allocation principle (*Equation 12*) shows that then not only the number of readout molecules needs to be raised but also the power. Clearly, improving the sensing precision comes at a cost: more copies of the components of the sensing system need to be synthesized every cell cycle, and more energy is needed to run the system. Our theory (i.e., *Equation 6*) makes it possible to derive the Pareto front that quantifies the trade-off between the maximal sensing precision and the cost of making the sensing system (see *Figure 5*). Importantly, the design of the optimal system at the Pareto front obeys, to a good approximation, our resource allocation principle (*Equation 12*). This is because this principle specifies the optimal *ratios* of $R_{\mathrm{T}}$, $X_{\mathrm{T}}$, $\dot{w}$, and $\tau_{\mathrm{r}}$ given the input statistics, and these ratios are fairly

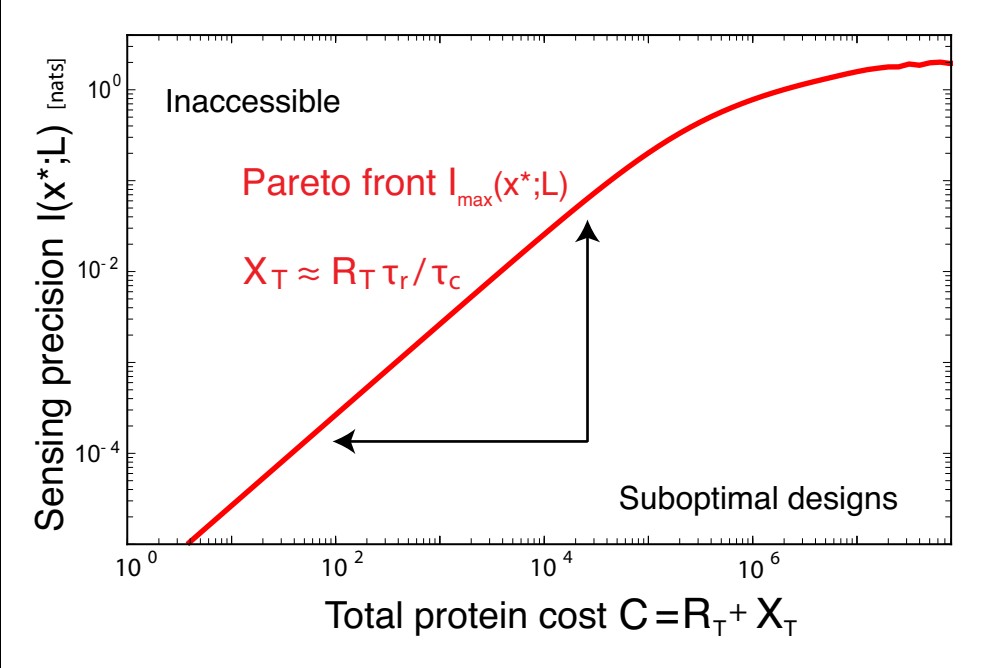

**Figure 5.** The benefit of a sensing system depends on the sensing precision it can achieve and the cost of making it. The Pareto front characterizes the trade-off between the maximal sensing precision, quantified by the maximal mutual information $I_{\mathrm{max}}(x^*;L)$, and the cost of making the sensing system, $C = R_{\mathrm{T}} + c_{\mathrm{X}}X_{\mathrm{T}}$, where $c_{\mathrm{X}}$ is the relative cost of making a readout versus a receptor protein, here taken to be $c_{\mathrm{X}} = 1$. System designs below the Pareto front are suboptimal and can be improved by reducing the cost, that is, the number of proteins, and / or increasing the sensing precision. The optimal systems at the Pareto front obey, to a good approximation, the allocation principle *Equation 12*. The Pareto front, formed by the maximal value $I_{\mathrm{max}}(x^*;L)$ of $I(x^*;L) = 1/2 \ln(1 + \mathrm{SNR})$ as a function of $C$, is obtained by minimizing *Equation 6* over $p, \tau_{\mathrm{r}}, R_{\mathrm{T}}, X_{\mathrm{T}}$ subject to the constraint $C = R_{\mathrm{T}} + X_{\mathrm{T}}$; the quality parameter is $q^{\mathrm{opt}} \approx 0.76$ corresponding to $\Delta\mu^{\mathrm{opt}} \approx 4 k_{\mathrm{B}}T$; $\tau_{\mathrm{c}}/\tau_{\mathrm{L}} = 10^{-2}$; $\sigma_L/\overline{L}_T = 10^{-2}$.

insensitive to the costs of the respective resources: resources that are in excess cannot improve sensing and are thus wasted, no matter how cheap they are. It probably explains why our theory, without any fit parameters, not only predicts the integration time that allows *E. coli* to sense shallow gradients (*Figure 4*) but also the number of receptor and readout molecules (*Govern and Ten Wolde, 2014a*).

In our study, we have limited ourselves to a canonical push–pull motif. Yet, the work of *Govern and Ten Wolde, 2014a* indicates that our results hold more generally, pertaining also to systems that employ cooperativity, negative or positive feedback, or multiple layers, as the MAPK cascade. While multiple layers and feedback change the response time, they do not make time integration more efficient in terms of readout molecules or energy (*Govern and Ten Wolde, 2014a*). And provided it does not increase the input correlation time (*Skoge et al., 2011*; *Ten Wolde et al., 2016*), cooperative ligand binding can reduce the sensing error per sample, but the resource requirements in terms of readout molecules and energy per sample do not change (*Govern and Ten Wolde, 2014a*). In all these systems, time integration requires that the history of the receptor is stored, which demands protein copies and energy.

Lastly, in this article we have studied the resource requirements for estimating the current concentration via the mechanism of time integration. However, to understand how *E. coli* navigates in a concentration gradient, we do not only have to understand how the system filters the high-frequency ligand-binding noise via time averaging but also how on longer timescales the system adapts to changes in the ligand concentration (*Sartori and Tu, 2011*). This adaptation system also exhibits a trade-off between accuracy, speed, and power (*Lan et al., 2012*; *Sartori and Tu, 2015*). Intriguingly, simulations indicate that the combination of sensing and adaptation allows *E. coli* not only to accurately estimate the current concentration but also the future ligand concentration (*Becker et al., 2015*). It will be interesting to see whether an optimal resource allocation principle can be formulated for systems that need to predict future ligand concentrations.

## Materials and methods

Methods are described in Appendices 1–3. Appendix 1 derives the central result of our article (*Equation 6*). Appendix 2 derives the fundamental resources and the corresponding sensing limits (*Equations 8 and 9*). Appendix 3 describes how the optimal gain and integration time can be understood using ideas from filtering and information theory.

## Acknowledgements

We wish to acknowledge Bela Mulder, Tom Shimizu, and Tom Ouldridge for many fruitful discussions and a careful reading of the manuscript. This work is part of the research program of the Netherlands Organisation for Scientific Research (NWO) and was performed at the research institute AMOLF.

## Additional information

### Funding

| Funder | Author |
| --- | --- |
| Nederlandse Organisatie voor Wetenschappelijk Onderzoek | Giulia Malaguti Pieter Rein ten Wolde |

The funders had no role in study design, data collection and interpretation, or the decision to submit the work for publication.

### Author contributions

Giulia Malaguti, Conceptualization, Investigation, Methodology, Writing - original draft, Writing - review and editing; Pieter Rein ten Wolde, Conceptualization, Resources, Funding acquisition, Investigation, Methodology, Writing - review and editing

## Author ORCIDs
Pieter Rein ten Wolde (iD) https://orcid.org/0000-0001-9933-4016

## Decision letter and Author response
Decision letter https://doi.org/10.7554/eLife.62574.sa1
Author response https://doi.org/10.7554/eLife.62574.sa2

## Additional files

### Supplementary files
• Transparent reporting form

### Data availability
All data generated or analysed during this study are included in the manuscript and supporting files.

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

## Appendix 1

### Signal-to-noise ratio

Here, we provide the derivation of the central result of this article, *Equation 6* of the main text. The derivation starts from the SNR, given in *Equation 2*. Here, $\sigma_L^2$ is the width of the input distribution, while $(\delta\hat{L})^2$ is the error in the estimate of the concentration. The latter is derived from the dynamic input–output relation $p_{\tau_r}(L)$, which is the mapping between the average receptor occupancy over the past integration time $\tau_r$ and the current ligand concentration $L$ (see *Figure 2*). Concretely, the error $(\delta\hat{L})^2$ is given by *Equation 1*, where $\sigma_{\hat{p}_{\tau_r}}^2$ is the error in the estimate of the average receptor occupancy over the past integration time $\tau_r$ and $\tilde{g}_{L\to p_{\tau_r}}$ is the dynamic gain, which is the slope of the dynamic input–output relation $p_{\tau_r}(L)$. Below, we first derive the dynamic gain $\tilde{g}_{L\to p_{\tau_r}}$ and then the error in the estimate of the receptor occupancy $\sigma_{\hat{p}_{\tau_r}}^2$.

### Dynamic input–output relation

The dynamic input–output relation $p_{\tau_r}(L)$ is the average receptor occupancy $p_{\tau_r}$ over the past integration time $\tau_r$, given that the current ligand concentration $L(t) = L$. The cell estimates $p_{\tau_r}$ via its receptor readout system, which is a device that takes samples of the receptor: the readout molecules at time $t$ constitute samples of the ligand-binding state of the receptor at earlier sampling times $t_i$ (see *Figure 2*). More specifically, the cell estimates $p_{\tau_r}$ from the number of active readout molecules $x^*(L(t)) = x^*(L)$:

$$\hat{p}_{\tau_r}(L) = \frac{x^*(L)}{\overline{N}}, \tag{14}$$

where $\overline{N}$ is the average of the number of samples $N$ taken during the integration time $\tau_r$. Hence, the dynamic input–output relation is

$$p_{\tau_r}(L) \equiv E\langle n(t_i)\rangle_{L(t)}, \tag{15}$$

where $n(t_i) = 0, 1$ is the receptor occupancy at time $t_i$, $E$ denotes the expectation over the sampling times $t_i$, and $\langle\ldots\rangle_{L(t)}$ denotes an average over receptor–ligand binding noise and the subensemble of ligand trajectories that each end at $L(t)$ (see *Figure 2c*); the quantity $\langle n(t_i)\rangle_{L(t)}$ is indeed the average receptor occupancy at time $t_i$, *given that the ligand concentration at time $t$ is $L(t) = L$*. Importantly, the receptor samples can also decay via the deactivation of $x^*$. Taking this into account, the probability that a readout molecule at time $t$ provides a sample of the receptor at an earlier time $t_i$ is $p(t_i|\text{sample}) = e^{-(t-t_i)/\tau_r}/\tau_r$ (*Govern and Ten Wolde, 2014a*). Averaging the receptor occupancy over the sampling times $t_i$ then yields

$$p_{\tau_r}(L) = \int_{-\infty}^{t} dt_i \langle n(t_i)\rangle_{L(t)} \frac{e^{-(t-t_i)/\tau_r}}{\tau_r}. \tag{16}$$

### Dynamic gain

When the current ligand concentration $L(t)$ deviates from its mean $\overline{L}$ by $\delta L(t) \equiv L(t) - \overline{L}$, then $p_{\tau_r}$ deviates on average from its mean $p$ (the average receptor occupancy over all $\delta L(t)$) by

$$\delta p_{\tau_r} \equiv p_{\tau_r} - p = E\langle\delta n(t_i)\rangle_{\delta L(t)} = \int_{-\infty}^{t} dt_i \langle\delta n(t_i)\rangle_{\delta L(t)} \frac{e^{-(t-t_i)/\tau_r}}{\tau_r}. \tag{17}$$

Here, $E$ denotes again the expectation over the sampling times $t_i$, and $\langle\delta n(t_i)\rangle_{\delta L(t)} \equiv \langle n(t_i)\rangle_{\delta L(t)} - p$ is the average deviation in the receptor occupancy $n(t_i)$ at time $t_i$ from its mean $p$, given that the ligand concentration at time $t$ is $\delta L(t)$ (see *Figure 2c*). We can compute it within the linear-noise approximation (*Gardiner, 2009*):

$$\langle \delta n(t_i) \rangle_{\delta L(t)} = \rho_n \int_{-\infty}^{t_i} dt' \langle \delta L(t') \rangle_{\delta L(t)} e^{-(t_i - t')/\tau_c}, \tag{18}$$

where $\rho_n = p(1-p)/(\overline{L}_T \tau_c)$ and $\langle \delta L(t') \rangle_{\delta L(t)}$ is the average ligand concentration at time $t'$, given that the ligand concentration at time $t$ is $\delta L(t)$. The latter is given by *Bowsher et al., 2013*

$$\langle \delta L(t') \rangle_{\delta L(t)} = \delta L(t) e^{-|t-t'|/\tau_L}. \tag{19}$$

Combining *Equations 17–19* yields the following expression for the average change in the average receptor occupancy $p_{\tau_r}$, given that the ligand at time $t$ is $\delta L(t)$:

$$\delta p_{\tau_r} = \frac{p(1-p)}{\overline{L}_T} \left(1 + \frac{\tau_c}{\tau_L}\right)^{-1} \left(1 + \frac{\tau_r}{\tau_L}\right)^{-1} \delta L(t), \tag{20}$$

$$\equiv \tilde{g}_{L \to p_{\tau_r}} \, \delta L(t). \tag{21}$$

Hence, the dynamic gain is

$$\tilde{g}_{L \to p_{\tau_r}} = \frac{p(1-p)}{\overline{L}} \left(1 + \frac{\tau_c}{\tau_L}\right)^{-1} \left(1 + \frac{\tau_r}{\tau_L}\right)^{-1}, \tag{22}$$

$$= g_{L \to p} \left(1 + \frac{\tau_c}{\tau_L}\right)^{-1} \left(1 + \frac{\tau_r}{\tau_L}\right)^{-1}. \tag{23}$$

The dynamic gain is the slope of the dynamic input–output relation $p_{\tau_r}(L)$ (see *Figure 2a*). It yields the average change in the receptor occupancy $p_{\tau_r}$ over the past integration time $\tau_r$ when the change in the ligand concentration at time $t$ is $\delta L(t)$. It depends on all the timescales in the problem and only reduces to the static gain $g_{L \to p} = p(1-p)/\overline{L}$ when the integration time $\tau_r$ and the receptor correlation time $\tau_c$ are both much shorter than the ligand correlation time $\tau_L$. The dynamic gain determines how much an error in the estimate of $p_{\tau_r}$ propagates to the estimate of $L(t)$.

## Error in receptor occupancy

We can derive the variance in the estimate of the receptor occupancy over the past integration time $\tau_r$, $\sigma_{\hat{p}_{\tau_r}}^2$, directly from *Equation 14* for the system in the irreversible limit (*Malaguti and Ten Wolde, 2019*). While this derivation is illuminating, it is also lengthy. For the fully reversible system studied here, we follow a simpler route. Since the average number of samples $\overline{N}$ over the integration time $\tau_r$ is constant, it follows from *Equation 14* that

$$\sigma_{\hat{p}_{\tau_r}}^2 = \frac{\sigma_{x^*|L}^2}{\overline{N}^2}, \tag{24}$$

where $\sigma_{x^*|L}^2$ is the variance in the number of phosphorylated readout molecules, conditioned on the signal at time $t$ being $L(t) = L$. The conditional variance (*Tostevin and ten Wolde, 2010*)

$$\sigma_{x^*|L}^2 = \sigma_{x^*}^2 - \tilde{g}_{L \to x^*}^2 \sigma_L^2 \tag{25}$$

is the full variance $\sigma_{x^*}^2$ of $x^*$ minus the variance $\tilde{g}_{L \to x^*}^2 \sigma_L^2$ that is due to the signal variations, given by the dynamic gain $\tilde{g}_{L \to x^*}$ from $L$ to $x^*$ times the signal variance $\sigma_L^2$.

The full variance of the readout $\sigma_{x^*}^2$ in *Equation 25* can be obtained from the linear-noise approximation (*Gardiner, 2009*), see *Malaguti and Ten Wolde, 2019*:

$$\sigma_{x^*}^2 = f(1-f)X_T + \frac{\rho'^2}{\mu'(\mu+\mu')}\left[p(1-p)R_T + \frac{\rho^2 \sigma_L^2(\lambda+\mu+\mu')}{\mu(\lambda+\mu)(\lambda+\mu')}\right]. \tag{26}$$

In this expression, $\mu = \tau_c^{-1} = k_1\overline{L} + k_2$ is the inverse of the receptor correlation time $\tau_c$; $p = \overline{RL}/R_T = k_1\overline{L}/(k_2 + k_1\overline{L}) = k_1\overline{L}\tau_c$ is the probability that a receptor is bound to ligand; $\rho = R_T k_1(1-p) = p(1-p)R_T\mu/\overline{L}$; $\mu' = \tau_r^{-1} = (k_f + k_{-f})pR_T + k_r + k_{-r}$ is the inverse of the integration time $\tau_r$; $f = \overline{x^*}/x_T = (k_f p R_T + k_{-r})\tau_r$ is the fraction of phosphorylated readout;

and $\rho' = k_f X_T(1-f) - k_{-f} X_T f = \dot{n}/(pR_T)$ is the total flux $\dot{n}$ around the cycle of readout activation and deactivation divided by the total number $pR_T$ of ligand-bound receptors: it is the rate at which each receptor is sampled, be it ligand bound or not. For what follows below, we note that the quality parameter $q = (e^{\Delta\mu_1} - 1)(e^{\Delta\mu_2} - 1)/(e^{\Delta\mu} - 1) = \rho' pR_T \tau_r/(f(1-f)X_T) = \dot{n}\tau_r/(f(1-f)X_T)$.

To get $\sigma^2_{\hat{p}_{\tau_r}}$ from *Equations 24 and 25,* we need not only $\sigma^2_x$ (*Equation 26*) but also the average number of samples $\overline{N}$ and the dynamic gain $\tilde{g}^2_{L \to x^*}$. The average number of samples taken during the integration time $\tau_r$ is $\overline{N} = \dot{n}\tau_r/p = f(1-f)X_T q/p = \rho' R_T/\mu'$, and the effective number of reliable samples is $\overline{N}_{eff} = q\overline{N}$. Since $p_{\tau_r}(L) = E\langle x^*\rangle_L/\overline{N}$, where $E\langle x^*\rangle_L$ is the average number of active readout molecules for a given input $L(t) = L$ and $\overline{N}$ is a constant independent of $L$, it follows that

$$\tilde{g}_{L \to x^*} = \tilde{g}_{L \to p_{\tau_r}}\overline{N} = \tilde{g}_{L \to p_{\tau_r}}R_T\frac{\rho'}{\mu'}, \tag{27}$$

with $\tilde{g}_{L \to p_{\tau_r}}$ the dynamic gain from $L$ to $p_{\tau_r}$, given by *Equation 22*. *Equation 27* can be verified via another route that does not rely on the sampling framework because we also know that $\tilde{g}_{L \to x^*} = \sigma^2_{L,x^*}/\sigma^2_L$ (*Tostevin and ten Wolde, 2010*), where the co-variance $\sigma^2_{L,x^*}$ can be obtained from the linear-noise approximation (*Malaguti and Ten Wolde, 2019*; *Gardiner, 2009*). Combining *Equations 24–27* yields

$$\sigma^2_{\hat{p}_{\tau_r}} = \frac{p(1-p)}{\overline{N}_{eff}} + \frac{p(1-p)}{R_T(1 + \tau_r/\tau_c)} + \frac{p^2}{\overline{N}_{eff}} + \tilde{g}^2_{L \to p_{\tau_r}}\sigma^2_L\left[\left(1 + \frac{\tau_c}{\tau_L}\right)\left(1 + \frac{\tau_r}{\tau_L}\right)\left(1 + \frac{\tau_c\tau_r}{\tau_L(\tau_c + \tau_r)}\right) - 1\right]. \tag{28}$$

This can be rewritten using the expression for the fraction of independent samples, which, assuming that $\tau_r \gg \tau_c$, is $f_I = 1/(1 + 2\tau_c/\Delta)$, with $\Delta = 2\tau_r R_T/\overline{N}_{eff}$ the effective spacing between the samples (*Govern and Ten Wolde, 2014a*):

$$\sigma^2_{\hat{p}_{\tau_r}} = \underbrace{\frac{p(1-p)}{f_I\overline{N}_{eff}} + \frac{p^2}{\overline{N}_{eff}}}_{\sigma^{2,\,samp}_{\hat{p}_{\tau_r}}} + \underbrace{\tilde{g}^2_{L \to p_{\tau_r}}\sigma^2_L\left[\left(1 + \frac{\tau_c}{\tau_L}\right)\left(1 + \frac{\tau_r}{\tau_L}\right)\left(1 + \frac{\tau_c\tau_r}{\tau_L(\tau_c + \tau_r)}\right) - 1\right]}_{\sigma^{2,\,dyn}_{\hat{p}_{\tau_r}}}, \tag{29}$$

Here, $\sigma^{2,\,samp}_{\hat{p}_{\tau_r}}$ is the sampling error in the estimate of $p_{\tau_r}$ (*Malaguti and Ten Wolde, 2019*); it is a statistical error, which arises from the finite cellular resources to sample the state of the receptor, protein copies, time, and energy (see *Figure 2b*). The other contribution, $\sigma^{2,\,dyn}_{\hat{p}_{\tau_r}}$, is the dynamical error in the estimate of $p_{\tau_r}$ (*Malaguti and Ten Wolde, 2019*); it is a systematic error that arises from the input dynamics and only depends on the average receptor occupancy and the timescales of the input, receptor, and readout (see *Figure 2c*); it neither depends on the number of protein copies nor on the energy necessary to sample the receptor.

## Final result: SNR

Combining *Equations 29* and *22* with *Equation 3* yields the principal result of our work (*Equation 6*) of the main text.

## Appendix 2

### Fundamental resources

To identify the fundamental resources limiting the sensing accuracy and derive the corresponding sensing limits (*Equations 8 and 9*), it is helpful to rewrite the SNR in terms of collective variables that illuminate the cellular resources. For that, we start from *Equation 6* of the main text and split the first term on the right-hand side and exploit the expression for the effective number of independent samples $\overline{N}_{\mathrm{I}} = 1/(1 + 2\tau_{\mathrm{c}}/\Delta)\overline{N}_{\mathrm{eff}}$ with $\Delta = 2\tau_{\mathrm{r}}R_{\mathrm{T}}/\overline{N}_{\mathrm{eff}}$. We then sum up the last two terms on the right-hand side and use that $\overline{N}_{\mathrm{eff}} = q\overline{N} = q\dot{n}\tau_{\mathrm{r}}/p$:

$$
\mathrm{SNR}^{-1} = \left(1 + \frac{\tau_{\mathrm{c}}}{\tau_{\mathrm{L}}}\right)^2 \left(1 + \frac{\tau_{\mathrm{r}}}{\tau_{\mathrm{L}}}\right)^2 \left[ \frac{(\overline{L}/\sigma_L)^2}{\overline{N}_{\mathrm{eff}} p(1-p)^2} + \frac{(\overline{L}/\sigma_L)^2}{p(1-p)R_{\mathrm{T}}(1 + \tau_{\mathrm{r}}/\tau_{\mathrm{c}})} \right]
$$
$$
+ \left(1 + \frac{\tau_{\mathrm{c}}}{\tau_{\mathrm{L}}}\right)\left(1 + \frac{\tau_{\mathrm{r}}}{\tau_{\mathrm{L}}}\right)\left(1 + \frac{\tau_{\mathrm{c}}\tau_{\mathrm{r}}}{\tau_{\mathrm{L}}(\tau_{\mathrm{c}} + \tau_{\mathrm{r}})}\right) - 1 \tag{30}
$$

$$
= \left(1 + \frac{\tau_{\mathrm{c}}}{\tau_{\mathrm{L}}}\right)^2 \left(1 + \frac{\tau_{\mathrm{r}}}{\tau_{\mathrm{L}}}\right)^2 \left[ \underbrace{\frac{(\overline{L}/\sigma_L)^2}{(1-p)^2 q\dot{n}\tau_{\mathrm{r}}}}_{\text{coding noise}} + \underbrace{\frac{(\overline{L}/\sigma_L)^2}{p(1-p)R_{\mathrm{T}}(1 + \tau_{\mathrm{r}}/\tau_{\mathrm{c}})}}_{\text{receptor input noise}} \right]
$$
$$
+ \underbrace{\left(1 + \frac{\tau_{\mathrm{c}}}{\tau_{\mathrm{L}}}\right)\left(1 + \frac{\tau_{\mathrm{r}}}{\tau_{\mathrm{L}}}\right)\left(1 + \frac{\tau_{\mathrm{c}}\tau_{\mathrm{r}}}{\tau_{\mathrm{L}}(\tau_{\mathrm{c}} + \tau_{\mathrm{r}})}\right) - 1}_{\text{dynamical error}}. \tag{31}
$$

The second term in between the square brackets describes the contribution to the sensing error that comes from the stochasticity in the concentration measurements at the receptor level. The first term in between the square brackets, the coding noise, describes the contribution that arises in storing these measurements into the readout molecules.

From *Equation 30,* the fundamental resources and the corresponding sensing limits (*Equations 8 and 9)* can be derived. Specifically, when the number of receptors and their integration are limiting, the coding noise in *Equation 30* is zero; exploiting that typically $\tau_{\mathrm{c}} \ll \tau_{\mathrm{r}}, \tau_{\mathrm{L}}$ and that the contribution to the sensing error from the receptor input noise is minimized for $p \to 1/2$, this yields *Equation 8* with $h = R_{\mathrm{T}}\tau_{\mathrm{r}}/\tau_{\mathrm{c}}$. When the number of readout molecules $X_{\mathrm{T}}$ is limiting, the receptor input noise is zero and $q \to 1$; noting that $\dot{n} = f(1-f)X_{\mathrm{T}}q/\tau_{\mathrm{r}}$ and that the contribution from the coding noise is minimized when $f \to 1/2$ and $p \to 0$, and again exploiting that $\tau_{\mathrm{c}} \ll \tau_{\mathrm{r}}, \tau_{\mathrm{L}}$, this yields *Equation 8* with $h = X_{\mathrm{T}}$. When the power $\dot{w} = \dot{n}\Delta\mu$ is limiting, then the receptor input noise is (again) zero. The coding noise is minimized for a given power constraint $\dot{w}$ when $\Delta\mu_1 = \Delta\mu_2 = \Delta\mu/2$, but two regimes can be distinguished based on the total free-energy drop $\Delta\mu$. When $\Delta\mu > 4k_{\mathrm{B}}T$, the system is in the irreversible regime and $q \to 1$ (see *Equation 7*); *Equation 30* shows that the error is then bounded by *Equation 8* with $h = \dot{w}\tau_{\mathrm{r}}/(\Delta\mu/4)$, using $\tau_{\mathrm{c}} \ll \tau_{\mathrm{r}}, \tau_{\mathrm{L}}$ and $p \to 0$. Yet, the sensing error is minimized in the quasi-equilibrium regime, where $\Delta\mu_1 = \Delta\mu_2 = \Delta\mu/2 \to 0$ and $q \to \beta\Delta\mu/4$, yielding *Equation 8* with $h = \beta\dot{w}\tau_{\mathrm{r}}$.

## Appendix 3

### The optimal gain and optimal integration time

The theory of the main text (*Equation 6*) is based on the idea that the cell uses its push–pull network to estimate the receptor occupancy $p_{\tau_r}(L)$ from which the current ligand concentration $L$ is then inferred by inverting the dynamic input–output relation $p_{\tau_r}(L)$. Yet, as we show here, this framework is equivalent to the idea that the cell estimates the concentration from the output $x^*$, using the dynamic input–output relation $x^*(L)$. Here, we use this observation to analyze our system using ideas from filtering and information theory. But first we demonstrate this correspondence.

To show that estimating the concentration from $\hat{p}_{\tau_r}$ is equivalent to that from estimating it from $x^*$, we first note that because the average number of samples $\overline{N}$ is constant, $\sigma_{x^*|L}^2 = \sigma_{\hat{p}_{\tau_r}}^2 \overline{N}^2$ while the gain from $L$ to $x^*$ is $\tilde{g}_{L\to x^*}^2 = \tilde{g}_{L\to p_{\tau_r}}^2 \overline{N}^2$. Consequently, the absolute error $(\delta\hat{L})^2$ in estimating the concentration via $x^*$, $(\delta\hat{L})^2 = \sigma_{x^*|L}^2/\tilde{g}_{L\to x^*}^2$, is the same as that of *Equation 1*: because the instantaneous number of active readout molecules $x^*$ reflects the average receptor occupancy $p_{\tau_r}$ over the past $\tau_r$, estimating the ligand concentration from $x^*$ is no different from inferring it from the average receptor occupancy $\hat{p}_{\tau_r} = x^*/\overline{N}$.

To make the connection with information and filtering theory, we note that in our Gaussian model the conditional distribution of $\delta x^*$ given $\delta L$ is given by *Tostevin and ten Wolde, 2010*

$$p(\delta x^*|\delta L) = \frac{1}{\sqrt{2\pi\sigma_{x^*|L}^2}} e^{-\frac{(\delta x^* - \tilde{g}_{L\to x^*}\delta L)^2}{2\sigma_{x^*|L}^2}}, \tag{32}$$

where $\tilde{g}_{L\to x^*}\delta L = \langle \delta x \rangle_L$ is the average value of $\delta x^*$ given that $\delta L(t) = \delta L$, and $\sigma_{x^*|L}^2$ is the variance of this distribution (see also *Equation 25*).

The relative error, the inverse of the SNR (see *Equation 2*), is

$$\text{SNR}^{-1} = \frac{(\delta\hat{L})^2}{\sigma_L^2} = \frac{\sigma_{x^*|L}^2}{\tilde{g}_{L\to x^*}^2 \sigma_L^2}. \tag{33}$$

As mentioned in the main text, the SNR also yields the mutual information $I(x^*;L) = 1/2\ln(1 + \text{SNR})$ between the input $L$ and output $x^*$ (*Tostevin and ten Wolde, 2010*).

The notion of an optimal integration time or optimal dynamic gain is well known from filtering and information theory (*Andrews et al., 2006*; *Hinczewski and Thirumalai, 2014*; *Becker et al., 2015*; *Monti et al., 2018a*; *Monti et al., 2018b*; *Mora and Nemenman, 2019*). To elucidate the optimal gain and integration time in our system, we combine the above equation with *Equations 25 and 26* to write the relative error as

$$\text{SNR}^{-1} = \underbrace{\frac{f(1-f)X_T}{\tilde{g}_{L\to x^*}^2 \sigma_L^2}}_{\text{readout switching noise}} + \underbrace{\frac{g_{RL\to x^*}^2 1/(1+\tau_r/\tau_c)p(1-p)R_T}{\tilde{g}_{L\to x^*}^2 \sigma_L^2}}_{\text{receptor input noise}}$$
$$+ \underbrace{\left(1+\frac{\tau_c}{\tau_L}\right)\left(1+\frac{\tau_r}{\tau_L}\right)\left(1+\frac{\tau_c\tau_r}{\tau_L(\tau_c+\tau_r)}\right) - 1}_{\text{dynamical error}}, \tag{34}$$

where $g_{RL\to x^*} = \rho'/\mu'$ is the static gain from $RL$ to $x^*$. Written in this form, the trade-offs in maximizing the mutual information $I(x^*;L)$ (and minimizing the relative error in estimating the concentration) become apparent: increasing the dynamic gain $\tilde{g}_{L\to x^*}$ by decreasing the integration time $\tau_r$ raises the slope of the input–output relation $x^*(L)$, which helps to lift the transmitted signal above the intrinsic binomial switching noise of the readout, $f(1-f)X_T$. Also, the dynamical error is minimized by minimizing $\tau_r$ and maximizing $\tilde{g}_{L\to x^*}$. Yet, for the second term, which describes how noise in the input signal arising from receptor switching, $p(1-p)R_T$, is propagated to the output $x^*$, there exists an optimal integration time that minimizes this term: while decreasing $\tau_r$ increases the dynamic gain, which helps to raise the signal above the noise, it also impedes time averaging of this switching noise, described by the factor $1/(1+\tau_r/\tau_c)$.

The mutual information is $I(x^*; L) = H(x^*) - H(x^*|L)$, with $H(x^*)$ the entropy of the marginal output distribution and $H(x^*|L)$ the entropy of the output distribution conditioned on the input. Hence, information theory shows that in the weak noise limit, information transmission is optimal when the entropy of the output distribution is maximized (*Laughlin, 1981*; *Tkacik et al., 2008*). Our system obeys this principle. Since the dynamic gain $\tilde{g}_{L\to x^*} = \rho\rho'\tau_{\mathrm{L}}{}^2\tau_{\mathrm{c}}\tau_{\mathrm{r}}/[(\tau_{\mathrm{c}} + \tau_{\mathrm{L}})(\tau_{\mathrm{r}} + \tau_{\mathrm{L}})] \propto R_{\mathrm{T}}X_{\mathrm{T}}$, the amplification of the signal rises with $R_{\mathrm{T}}$ and $X_{\mathrm{T}}$. Since the standard deviation of the noise added to the transmitted signal coming from the stochastic receptor and readout activation scales with $\sqrt{R_{\mathrm{T}}}$ and $\sqrt{X_{\mathrm{T}}}$, respectively, it is clear that the *SNR* increases with $\sqrt{R_{\mathrm{T}}}$ and $\sqrt{X_{\mathrm{T}}}$. In the limit that $R_{\mathrm{T}}, X_{\mathrm{T}} \to \infty$, the relative error $\mathrm{SNR}^{-1}$ is only set by the dynamical error, which can be reduced to zero by $\tau_{\mathrm{r}} \to 0$, exploiting that typically $\tau_{\mathrm{c}} \ll \tau_{\mathrm{L}}$. This is the weak-noise limit in which the mutual information $I(x^*; L)$ is maximized by maximizing the entropy of the output distribution $H(x^*)$. Indeed, $\tau_{\mathrm{r}} \to 0$ corresponds to maximizing the gain, which maximizes the width of the output distribution, in this limit equal to $\sigma_x^2 = \tilde{g}_{L\to x^*}^2\sigma_L^2$ (see *Equation 25*), and thereby the entropy of the output distribution $H(x^*) = 1/2\ln(2\pi e\sigma_x^2)$.

Finally, we note that our Gaussian model is linear such that the central control parameter, besides protein copies and energy, is the integration time or the dynamic gain, which sets the slope of the linear input–output relation. While Wiener–Kolmogorov and Kalman filtering are exact only for these Gaussian models, information theory also applies to non-linear systems with non-Gaussian statistics. It has been used to show that neuronal systems (*Laughlin, 1981*; *Brenner et al., 2000*; *Fairhall et al., 2001*; *Nemenman et al., 2008*; *Tkacik et al., 2010*), signaling and gene networks (*Segall et al., 1986*; *Tkacik et al., 2008*; *Tkačik and Walczak, 2011*; *Nemenman, 2012*; *Dubuis et al., 2013*), and circadian systems (*Monti and Wolde, 2016*; *Monti et al., 2018a*) can maximize information transmission by optimizing the shape of the input–output relation (*Laughlin, 1981*; *Brenner et al., 2000*; *Fairhall et al., 2001*; *Tkacik et al., 2008*; *Monti et al., 2018a*); by desensitization, that is, adapting the output to the mean input via incoherent feedforward or negative feedback (*Segall et al., 1986*); by gain control, that is, adapting the output to the variance of the input by capitalizing on a steep response function and temporal correlations in the input (*Nemenman, 2012*); by removing coding redundancy via temporal decorrelation (*Nemenman et al., 2008*); by optimizing the tiling of the output space via the topology of the network (*Tkačik and Walczak, 2011*; *Dubuis et al., 2013*); or by exploiting cross-correlations between the signals (*Tkacik et al., 2010*; *Monti and Wolde, 2016*).

