## [Decision Letter]

**Acceptance summary:**

Malaguti and ten Wolde combine analysis of sensing limits for time-varying signals with previous work on resource allocation to minimize the costs of sensing. The work makes a solid contribution towards understanding the principles of resource allocation, and the conclusion that *E. coli* chemotaxis is optimized for shallow gradients should stimulate further discussion and work.

**Decision letter after peer review:**

Thank you for submitting your article "Theory for the optimal detection of time-varying signals in cellular sensing systems" for consideration by *eLife*. Your article has been reviewed by two peer reviewers, one of whom is a member of our Board of Reviewing Editors, and the evaluation has been overseen by Detlef Weigel as the Senior Editor. The reviewers have opted to remain anonymous.

The reviewers have discussed the reviews with one another and the Reviewing Editor has drafted this decision to help you prepare a revised submission.

Summary:

Malaguti and ten Wolde combine analysis of Mora-Nemenman (PRL 2019) on Berg-Purcell type sensing limits for time-varying signals with previous work of one of the authors (ten Wolde, PNAS 2014) on resource allocation to minimize the costs of sensing. The previous work discussed tradeoffs for the costs of sensing a constant signal. Here it is extended to time-varying signals (modeled as colored-noise Gaussian). Although the authors exaggerate the novelty of their analysis, they make a solid contribution towards understanding the principles of resource allocation. The conclusion that *E. coli* chemotaxis is optimized for shallow gradients seems reasonable and should stimulate further discussion and work. Despite flaws in establishing context, the paper deserves publication

While enthusiastic about the results in the manuscript, the reviewers found that there are issues involving claims of novelty, and the overall presentation of mathematical results that will need considerable revision.

1) The novelty of their analysis is exaggerated.

The basic problem of optimally estimating the state of a time-varying signal probed by noisy measurements is textbook material for engineers. Here, because everything is linearized about an operating point and because noise is approximated as Gaussian, there is an exact analytic theory going back to Kalman and Bucy in 1961. The result underlies the Mora-Nemenman analysis (although they, too, did not reference that work for the Gaussian approximation used for their concrete results) and that done here. In engineering, the general problem optimally estimating a stochastic signal via noisy measurements was already considered by Kolmogorov and Wiener in the 1940s and formulated in the time domain, for linear systems driven by white noise (the approximation used here) by Kalman and Bucy.

It is a weakness that the connections are not mentioned specifically by referencing. Appealing to known results not only underlines how different disciplines often need to tackle the same problems, it allows for the use of textbook results and can shorten a paper. As a corollary, authors who map their problem onto known results should not be penalized for doing so, when the application is new and important (as here).

2) Specific sentences have language that seems exaggerated, in the light of historical work on filtering theory:

"Our theory is based on a new concept, the dynamic input-output relation p_τr_(L)."

- Dynamical models (with dynamical input-output relations) are a central element of filtering theory.

".Our theory reveals that the sensing error can be decomposed into two terms [sampling (sensing) error and dynamical error]."

- The framework of filtering theory assumes noisy measurements of stochastic signals.

"Our theory illuminates how the optimal design depends on the timescale of the input τ_L_."

- The statement is true but should be framed in the context of other work, including in systems biology / biophysics, which comes to the same conclusion. For example, the work of Laughlin in 1981, and later Nemenman and Bialek, show that the input-output "gain function" should be adapted to the statistics of the input (including time scales of variation). Bialek's 2012 Biophysics book discusses many examples.

3) In its present form, the paper will be essentially unreadable by the vast majority of the *eLife* readership; the writing style is more appropriate to the Physical Review than to *eLife*. Indeed, much of the paper is written like a technical note that builds on previous work (Berg/Purcell, etc.) without sufficient explanation for the general reader. This criticism extends to the explanations of the underlying model, the lack of definition of terminology, and the notation.

Here are but a few examples of the points above. The authors are encouraged to rethink the entire presentation de novo for maximum improvement.

a) The definition of a "push-pull" network should be given in the paper.

b) For the purpose of readability by a diverse audience some care should be taken to define technical terms explicitly (e.g. "Markovian"), and to explain various statements more clearly. Appearing so early in the paper, such condensed statements will be off-putting to the general reader.

c) The whole notation of "inverting the mapping" that is central to the theory is not well-explained.

d) Regarding notation: consider, for example, the caption of Figure 2 and Equation 1, in which there is the quantity σ2p^tr|L . This is simply too complicated, and its meaning will be utterly opaque to the general reader.

4) In many ways it seems that if the authors presaged the development of the theory by indicating the run-and-tumble context early on it would help the reader to understand the precise motivations behind their lengthy calculations, and to give an idea of the time scales.

---

## [Author Response]

Revisions for this paper:While enthusiastic about the results in the manuscript, the reviewers found that there are issues involving claims of novelty, and the overall presentation of mathematical results that will need considerable revision.1) The novelty of their analysis is exaggerated.The basic problem of optimally estimating the state of a time-varying signal probed by noisy measurements is textbook material for engineers. Here, because everything is linearized about an operating point and because noise is approximated as Gaussian, there is an exact analytic theory going back to Kalman and Bucy in 1961. The result underlies the Mora-Nemenman analysis (although they, too, did not reference that work for the Gaussian approximation used for their concrete results) and that done here. In engineering, the general problem optimally estimating a stochastic signal via noisy measurements was already considered by Kolmogorov and Wiener in the 1940s and formulated in the time domain, for linear systems driven by white noise (the approximation used here) by Kalman and Bucy.It is a weakness that the connections are not mentioned specifically by referencing. Appealing to known results not only underlines how different disciplines often need to tackle the same problems, it allows for the use of textbook results and can shorten a paper. As a corollary, authors who map their problem onto known results should not be penalized for doing so, when the application is new and important (as here).

We fully acknowledge the point that is made here and we agree that we should have described the connection with filtering theory much more clearly, especially given the fact that we have applied filtering theory ourselves to biochemical signalling before (Becker, Mugler, Ten Wolde, 2015).

We indeed employ a Gaussian model, and for such a Gaussian model Kalman and Wiener-Kolmogorov filtering theory become exact, as the reviewers correctly point out. In our previous work, we studied different types of input signals (Markovian and non-Markovian), and then used Wiener-Kolmogorov filtering theory to derive the optimal topology of the sensing network for the different types of input signals (Becker et al., 2015). In our current work, we consider one type of input signal—a stationary Markovian signal. The optimal filter for this type of input signal is a low-pass, exponential filter and the canonical network motif studied in our current study implements precisely such a filter (see Becker et al., 2015). This filter is useful, because it allows the system to time integrate the input signal and hence filter out the high-frequency input noise arising here from receptor-ligand binding. We agree with the reviewers that these ideas are well established and that we should describe them more clearly in our manuscript.

Yet, while filtering theory provides a powerful approach to elucidating the optimal topology and response dynamics of the sensing network, as determined by the statistics of the input signal, it does not naturally reveal the resource requirements for sensing. To this end, we have generalised the sampling framework that we have developed previously (Govern and Ten Wolde, 2014). This framework is particularly well suited for elucidating the resource requirements for time integration, because it starts from the observation that the receptor-readout system consists of discrete molecules that interact with the receptor in a stochastic fashion: our description is thus based on the idea that the readout system is a sampling device that implements the mechanism of time integration not by continuously integrating the state of the receptor, but rather by discretely and stochastically sampling it, via the collisions of the readout molecules with the receptor proteins. The novelty of this manuscript is that we have now extended this sampling framework, for the first time, to time-varying signals.

In the revised manuscript, we have thoroughly rewritten the Introduction and added two new paragraphs to the Discussion. In the Introduction, we now mention that the problem of optimally predicting time-varying signals is a classic problem, for which different theories have been developed. We then also briefly review the application of filtering theories to biochemical signalling, adding relevant references. We then emphasise that we consider one class of input signals and one canonical network motif, which is known to implement a filter that is optimal for this type of signals, and raise the central question of our manuscript, which is what the cellular resource requirements are for optimally implementing such a network. We end the Introduction with a brief discussion of the main ideas of our theory and our findings.

In the revised Introduction, we also note that our theory applies to systems that employ the mechanism of time integration, and not to systems that employ Maximum-Likelihood sensing or Bayesian filtering, which underlies the analysis of Mora and Nemenman, 2019.

In the new paragraphs in the Discussion section, we first emphasise that our model is Gaussian and that the performance criterion of our theory, minimizing the mean-square error, is identical to that of Wiener and Kalman filtering theory, which are exact for Gaussian models. We then mention that for our Gaussian model minimizing the mean-square error is equivalent to maximizing the mutual information, thus making a connection between our theory, filtering theories, and information theory. In the next paragraph, we then discuss the concepts that have emerged from these theories and we describe how our findings relate to these concepts; here we have also added a new Appendix 3, where we work out this connection in more mathematical detail.

We thank the reviewers and the editor for this important comment, because we agree that addressing it puts our work into a broader context.

2) Specific sentences have language that seems exaggerated, in the light of historical work on filtering theory:"Our theory is based on a new concept, the dynamic input-output relation p_τr_(L)."- Dynamical models (with dynamical input-output relations) are a central element of filtering theory.

We fully agree that dynamic input-output relations are a key concept in filtering theory and we acknowledge that similar input-output relations have been studied before in the biophysics literature (Bialek et al., IEEE, 2006; Tostevin and Ten Wolde, 2010; Nemenman, 2012; Hinczewski and Thirumalai, 2014; Becker et al., 2015). We merely wanted to emphasise that our dynamic input-output relation *p_τr_*(*L*) differs fundamentally from the conventional static input-output relations that are typically considered in the context of sensing static signals (Berg and Purcell, 1977; Bialek and Setayesghar, 2005; Kaizu et al., 2014; Mugler et al., 2016).

We have completely rewritten this paragraph. As we describe in our response to the previous point, we now first introduce filtering theory, then describe the novel aspect of the current manuscript, which is the extension of our sampling framework to time-varying signals, and then mention the dynamic input-output relation, contrasting it with the static input-output relation used to describe the sensing of static signals.

"Our theory reveals that the sensing error can be decomposed into two terms [sampling (sensing) error and dynamical error]."- The framework of filtering theory assumes noisy measurements of stochastic signals.

We agree with this, but our decomposition is more specific, showing how the sampling error and the dynamical error depend on the cellular resources protein copies, time, and energy. Our decomposition identifies the combinations of cellular resources, the “collective variables”, that control the sensing precision. It is this decomposition that allows us to make specific predictions on the optimal design of sensing systems that maximize the sensing precision given resource constraints. This is the novel contribution of our paper.

We now emphasise this more clearly.

"Our theory illuminates how the optimal design depends on the timescale of the input τ_L_."- The statement is true but should be framed in the context of other work, including in systems biology / biophysics, which comes to the same conclusion. For example, the work of Laughlin in 1981, and later Nemenman and Bialek, show that the input-output "gain function" should be adapted to the statistics of the input (including time scales of variation). Bialek's 2012 Biophysics book discusses many examples.

The idea of an optimal integration time, and the related ideas of matching the gain and the input-output relation to the statistics of the input, are indeed well known. In fact, we had cited a number of papers that also show the existence of an optimal integration time (Becker et al., 2015; Monti et al., 2018; Mora and Nemenman, 2019). The aim of our paragraph in the Discussion, was to discuss how the input timescale governs the optimal design in terms of the cellular resources receptors, readout molecules, and energy—this is the novel aspect of our work. But we agree with the reviewers that it is important to connect our observations to previous ideas on the optimal gain and input-output relations that are matched to the statistics of the input. We therefore now first discuss here, in the Discussion section, these ideas and then describe how our findings relate to them. After we have discussed this connection, we emphasise that our sampling framework gives a detailed description of the optimal design in terms of the required cellular resources, and then discuss how this depends on the statistics of the input signal.

More concretely, we first point out that our model is Gaussian and that the performance criterion of our theory is minimizing the mean-squared error, which is precisely the performance criterion of Wiener and Kalman filtering. We emphasise that the push-pull network is an exponential filter and that this is the optimal filter for the memoryless (Markovian) signals as studied here. We also point out that for our Gaussian model minimizing the mean-square error is equivalent to maximizing the mutual information, thus making the connection between our work, filtering theory, and information theory. In the next paragraph, we then describe concepts that have emerged from filtering and information theory, and describe how our findings relate to these concepts. Here, we also refer to a new Appendix 3, in which we work this connection out in more (mathematical) detail. In this appendix we also discuss concepts for optimizing information transmission in non-linear systems, which are beyond the scope of the current model.

3) In its present form, the paper will be essentially unreadable by the vast majority of the eLife readership; the writing style is more appropriate to the Physical Review than to eLife. Indeed, much of the paper is written like a technical note that builds on previous work (Berg/Purcell, etc.) without sufficient explanation for the general reader. This criticism extends to the explanations of the underlying model, the lack of definition of terminology, and the notation.Here are but a few examples of the points above. The authors are encouraged to rethink the entire presentation de novo for maximum improvement.

We acknowledge that the audience of *eLife* is broader than that of Physical Review and we have rewritten the manuscript drastically. We now try to avoid technical terms like “Markovian”, “mapping”, “push-pull” network as much as possible, and where this cannot be avoided we have explained them for a broader audience. Indeed, we have carefully verified whether *all* the necessary concepts that are needed to follow our analysis are explained for a broad audience. For example, in the description of our model and our theory we now describe in more detail the concept of the input correlation time, receptor correlation time, readout relaxation time, the idea of time integration via discrete sampling of the receptor state. We have also simplified notation. Most importantly, perhaps, we have tried to rewrite the manuscript in a more lucid style so that it is easier to follow for a broad audience (see, for example, the description of the optimal design principle, Equation 12). With these improvements, we strongly believe that the revised manuscript is accessible for the broad readership of *eLife*, also because the previous, well cited, paper on which the current manuscript builds, is written with a similar level of detail and published in a journal with a similarly broad audience (Govern and Ten Wolde, 2014).

a) The definition of a "push-pull" network should be given in the paper.

While we could replace “push-pull network” with “cellular network” or simply “network”, we think it does help to keep this term, because it specifically refers to the cycle of readout phosphorylation and dephosphorylation downstream of the receptor. We have therefore decided to keep this term. We do, however, now explain it in the Introduction when we introduce it for the first time.

b) For the purpose of readability by a diverse audience some care should be taken to define technical terms explicitly (e.g. "Markovian"), and to explain various statements more clearly. Appearing so early in the paper, such condensed statements will be off-putting to the general reader.

Since “Markovian” only appears twice in the manuscript, we have changed “Markovian” by “memoryless” with an explanation. The concepts of “correlation time”, “integration time”, and “relaxation time”, are important and used widely in the manuscript; we have therefore added explanations the first time we introduce these, see Section “Theory: Model”. We have also clarified other technical terms, such as time averaging, sampling device, coding.

c) The whole notation of "inverting the mapping" that is central to the theory is not well-explained.

We have clarified that.

d) Regarding notation: consider, for example, the caption of Figure 2 and Equation 1, in which there is the quantity σ2p^tr|L This is simply too complicated, and its meaning will be utterly opaque to the general reader.

We have now simplified the notation as much as possible, while trying to keep clear to what specific quantity the symbol actually refers. For example, we have simplified σ2p^tr|L to σ2p^tr; we needed to retain the subscript *τ_r_*because this quantity denotes the variance not of the receptor occupancy *p* but rather that of the receptor occupancy *p_τr_*over the integration time *τ_r_*.

4) In many ways it seems that if the authors presaged the development of the theory by indicating the run-and-tumble context early on it would help the reader to understand the precise motivations behind their lengthy calculations, and to give an idea of the time scales.

We would like to emphasise that the analysis is much more generic than the specific application to the *E. coli* chemotaxis system. We therefore would like to maintain the distinction between the general theory and the application of it to *E. coli* chemotaxis. However, we also agree that introducing this system earlier helps to understand the motivation and main ideas behind our theory. We have therefore expanded the discussion of this system in the Introduction.